# Observation of Rabi dynamics with a short-wavelength free-electron laser

Saikat Nandi[1✉], Edvin Olofsson[2], Mattias Bertolino[2], Stefanos Carlström[2], Felipe Zapata[2], David Busto[2], Carlo Callegari[3], Michele Di Fraia[3], Per Eng-Johnsson[2], Raimund Feifel[4], Guillaume Gallician[5], Mathieu Gisselbrecht[2], Sylvain Maclot[2,4], Lana Neoričić[2], Jasper Peschel[2], Oksana Plekan[3], Kevin C. Prince[3], Richard J. Squibb[4], Shiyang Zhong[2], Philipp V. Demekhin[6], Michael Meyer[7], Catalin Miron[5,8], Laura Badano[3], Miltcho B. Danailov[3], Luca Giannessi[3,9], Michele Manfredda[3], Filippo Sottocorona[3,10], Marco Zangrando[3,11] & Jan Marcus Dahlström[2✉]

Rabi oscillations are periodic modulations of populations in two-level systems interacting with a time-varying field[1]. They are ubiquitous in physics with applications in different areas such as photonics[2], nano-electronics[3], electron microscopy[4] and quantum information[5]. While the theory developed by Rabi was intended for fermions in gyrating magnetic fields, Autler and Townes realized that it could also be used to describe coherent light–matter interactions within the rotating-wave approximation[6]. Although intense nanometre-wavelength light sources have been available for more than a decade[7–9], Rabi dynamics at such short wavelengths has not been directly observed. Here we show that femtosecond extreme-ultraviolet pulses from a seeded free-electron laser[10] can drive Rabi dynamics between the ground state and an excited state in helium atoms. The measured photoelectron signal reveals an Autler–Townes doublet and an avoided crossing, phenomena that are both fundamental to coherent atom–field interactions[11]. Using an analytical model derived from perturbation theory on top of the Rabi model, we find that the ultrafast build-up of the doublet structure carries the signature of a quantum interference effect between resonant and non-resonant photoionization pathways. Given the recent availability of intense attosecond[12] and few-femtosecond[13] extreme-ultraviolet pulses, our results unfold opportunities to carry out ultrafast manipulation of coherent processes at short wavelengths using free-electron lasers.

The advent of free-electron laser (FEL) facilities, providing femtosecond light pulses in the gigawatt regime at extreme-ultraviolet (XUV) or X-ray wavelengths, has opened up many prospects for experiments in isolated atoms and molecules in the gas phase[14,15]. Over the past decade, pioneering results concerning multi-photon ionization of atoms[16] and small molecules[17] have been obtained using pulses from self-amplified spontaneous emission (SASE) FEL sources[8]. However, these pulses are prone to a low degree of coherence and poor shot-to-shot reproducibility owing to the instability inherent to the SASE process. As a result, despite theoretical predictions to observe Rabi dynamics at short wavelengths[18–21], its effects on the measured spectra were only indirect[22,23]. Instead, XUV pulses from a SASE FEL have been used as a pump that allowed subsequent ultrafast Rabi dynamics to be driven by laser pulses at near-infrared wavelengths[24]. In this regard, XUV pulses from a seeded FEL, such as FERMI (Free Electron laser Radiation for Multidisciplinary Investigations)[10], with its high temporal and spatial coherence, and large peak intensity can allow the study of coherent light–matter interactions[25] and phase-dependent interference effects of the wavefunction[2].

According to the Rabi model[1], if a two-level atom initially in its ground state $|a\rangle$, is subjected to an interaction with a field of frequency $\omega$ that couples it to the excited state $|b\rangle$, the probability for excitation varies sinusoidally in time as $P_b(t) = |\frac{\Omega}{W}|^2 \sin^2\left(\frac{Wt}{2}\right)$. The oscillating population leads to a symmetric structure in the frequency domain, known as an Autler–Townes (AT) doublet. The splitting is given by the generalized Rabi frequency $W = \sqrt{\Omega^2 + \Delta\omega^2}$, where $\Delta\omega = \omega - \omega_{ba}$, is the detuning of the field with respect to the atomic transition frequency, $\omega_{ba}$. The Rabi frequency for light–matter interaction within the dipole approximation is $\Omega = eE_0 z_{ba}/\hbar$, with $E_0$ being the electric field amplitude, $z_{ba}$ the transition matrix element, $\hbar$ the reduced Planck constant and $e$ the elementary charge. In addition to the periodic population transfer $P_b(t)$, the coherent dynamics is further associated with sign changes of the oscillating amplitudes for the two states. For fermions, such sign changes of the wavefunction can be connected to rotations in real space[1] that

[1]Université de Lyon, Université Claude Bernard Lyon 1, CNRS, Institut Lumière Matière, Villeurbanne, France. [2]Department of Physics, Lund University, Lund, Sweden. [3]Elettra-Sincrotrone Trieste, Trieste, Italy. [4]Department of Physics, University of Gothenburg, Gothenburg, Sweden. [5]Université Paris-Saclay, CEA, CNRS, LIDYL, Gif-sur-Yvette, France. [6]Institute of Physics and CINSaT, University of Kassel, Kassel, Germany. [7]European XFEL, Schenefeld, Germany. [8]ELI-NP, "Horia Hulubei" National Institute for Physics and Nuclear Engineering, Magurele, Romania. [9]Istituto Nazionale di Fisica Nucleare, Laboratori Nazionali di Frascati, Frascati, Italy. [10]Università degli Studi di Trieste, Trieste, Italy. [11]IOM-CNR, Istituto Officina dei Materiali, Trieste, Italy. ✉e-mail: saikat.nandi@univ-lyon1.fr; marcus.dahlstrom@matfys.lth.se

have been measured for neutron beams in magnetic fields[26]. Analogous sign changes in quantum optics were studied using Rydberg atoms to determine the number of photons in a cavity[27]. Recently, the sign changes in Rabi amplitudes have been predicted to strongly alter AT doublet structures in photo-excited atoms, when probed by attosecond XUV pulses[28].

Here we investigate the Rabi dynamics at XUV wavelengths in helium atoms induced by an intense pulse from the FERMI seeded FEL that couples the two levels $|a\rangle = 1s^2$ ($^1S_0$) and $|b\rangle = 1s4p$ ($^1P_1$), with $\hbar\omega_{ba} = \epsilon_b - \epsilon_a = 23.742$ eV (ref. [29]). The term $\epsilon_a$ ($\epsilon_b$) denotes the energy of the state $|a\rangle$ ($|b\rangle$). The dynamics is probed in situ by recording photoelectrons ejected from the state $|b\rangle$ or $|a\rangle$ during the ultrashort interaction, with one or two XUV-FEL photons, as illustrated in Fig. 1a. To interpret this nonlinear dynamics, we have developed an analytical model by partitioning the Hilbert space into the two-level system and its complement. We expand the time evolution in the form of a Dyson series, where in the zeroth order, the two-level system undergoes Rabi oscillations. The photoionization dynamics from the excited (ground) state is treated by first (second)-order time-dependent perturbation theory describing the FEL interaction with the complement of the two-level system (see Supplementary Information for details). The resulting AT doublet structure depends on whether the photoelectron is originating from the ground state, $|a\rangle$, or the excited state, $|b\rangle$, as shown in Fig. 1b. The narrow spectral bandwidth of the XUV-FEL pulse (20–65 meV; Methods) enables efficient coupling of $|a\rangle$ and $|b\rangle$ with the dipole element $z_{ba} = 0.1318a_0$, with $a_0$ being the Bohr radius[30]. As shown in Fig. 1c, several physical effects can impede the duration of Rabi cycling: spontaneous emission sets a fundamental limit at about 4 ns (ref. [31]), whereas one-photon ionization from the excited state (I) and two-photon non-resonant ionization from the ground state (II) restrict Rabi cycling at progressively higher intensities. The present experiment (diamond) can be driven coherently, as it takes place over an ultrafast time duration, with the estimated full-width at half-maximum (FWHM) of the driving FEL pulse duration being 56 ± 13 fs (ref. [32]). This is three orders of magnitude shorter than the lifetime for photoionization of the Rabi cycling atom $\tau_{a+b} \approx 100$ ps (see Supplementary Information for details). Thus, the Hamiltonian for a two-level system $H = \frac{1}{2}\hbar\omega_{ba}\hat{\sigma}_z + \hbar\Omega\cos(\omega t)\hat{\sigma}_x$, where $\hat{\sigma}_z$ and $\hat{\sigma}_x$ are Pauli operators, can be satisfied by a time-dependent wavefunction of the form $|\Psi(t)\rangle = a(t)e^{-i\epsilon_a t/\hbar}|a\rangle + b(t)e^{-i\epsilon_b t/\hbar}|b\rangle$. Within the rotating-wave approximation, the amplitudes of the ground and excited states are expressed as:

$$\begin{cases} a(t) = \left[\cos\dfrac{Wt}{2} - i\dfrac{\Delta\omega}{W}\sin\dfrac{Wt}{2}\right]\exp(i\Delta\omega t/2) \\ b(t) = -i\dfrac{\Omega}{W}\sin\dfrac{Wt}{2}\exp(-i\Delta\omega t/2), \end{cases} \quad (1)$$

provided that the electric field can be approximated as a flat-top shape in time. The sign changes associated with these Rabi amplitudes are essential to understand the ultrafast build-up of AT doublets from $|a\rangle$ or $|b\rangle$, by absorption of two or one resonant XUV-FEL photons. The AT doublet emerges owing to a destructive interference effect between photoelectrons ejected before and after the first sign change, which is found to occur at 1/2 and 1 Rabi period for the amplitudes $a(t)$ and $b(t)$ with $\Delta\omega = 0$, respectively. This is in agreement with the results from the analytical model presented in Fig. 1b. The contribution from the excited state, $|b\rangle$, can be understood as the Fourier transform of the time-dependent amplitude $b(t)$. Similarly, the contribution from the ground state, $|a\rangle$, can be related to the Fourier transform of $a(t)$ through a non-resonant wave packet composed of complement states, $|c\rangle$, as illustrated in Fig. 1a. Thus, the observation of a doublet from either state $|a\rangle$ or state $|b\rangle$ is a direct measurement of Rabi dynamics in the energy domain.

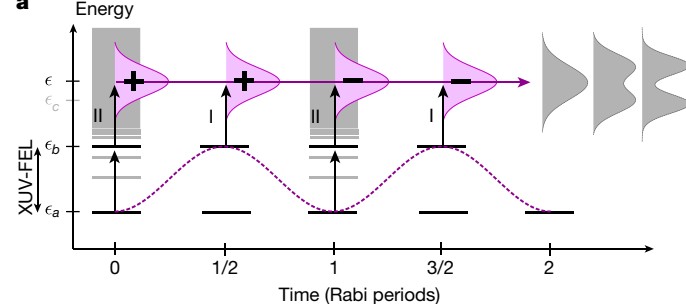

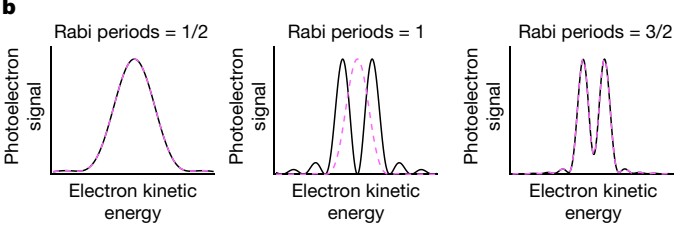

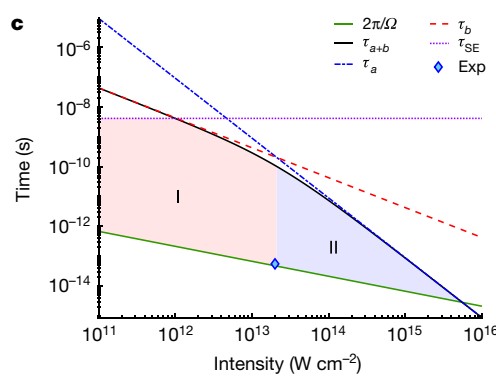

**Fig. 1 | Rabi oscillations induced by an XUV-FEL pulse. a**, The sinusoidal energy transfer between the XUV-FEL coupled states $|a\rangle$ and $|b\rangle$ (black horizontal lines) is associated with sign changes of state amplitudes between adjacent Rabi cycles (+ and −). Photoelectrons can be ejected from excited state $|b\rangle$, by one (I) photon, or by two (II) photons from $|a\rangle$ through complement states, $|c\rangle$ (grey horizontal lines). This results in the time-dependent build-up of an ultrafast AT doublet structure. **b**, The build-up of an ultrafast AT doublet for 1/2, 1 and 3/2 completed Rabi periods is shown for I-photon ionization from $|b\rangle$ (dashed, magenta line) and II-photon ionization from $|a\rangle$ (solid, black line) using the analytical model described in the Supplementary Information. **c**, Domains of photoionization from $1s^2$–$4p$ Rabi cycling helium atoms with the dominant photon process: I (red shaded area) and II (blue shaded area). Rabi cycling is limited by spontaneous emission ($\tau_{SE}$), I-photon ionization from $|b\rangle$ ($\tau_b$) and II-photon ionization from $|a\rangle$ ($\tau_a$) at progressively higher intensities of the FEL field. $\tau_{a+b}$ is the lifetime of the Rabi cycling atom subject to photoionization. The boundary between the two domains is determined by $\tau_a = \tau_b$. The diamond marks the experiment: on the boundary between the I and II domains and close to a single Rabi cycle.

Measured photoelectron spectra, shown in Fig. 2a, exhibit an AT splitting of $\hbar\Omega = 80 \pm 2$ meV (see Methods for details about the blind deconvolution procedure used here; the reported uncertainty is obtained from a fit of the symmetric AT doublet with two Voigt profiles with the same width). The corresponding Rabi period $2\pi/\Omega \approx 52$ fs, given its proximity to the FWHM of the XUV-FEL pulse, suggests that the experiment was performed in a regime of ultrafast AT doublet formation close to a single Rabi cycle. A slight blue detuning of the XUV light, by about 11 meV relative to the atomic transition[29], is required to record a symmetric AT doublet (black squares in Fig. 2a).

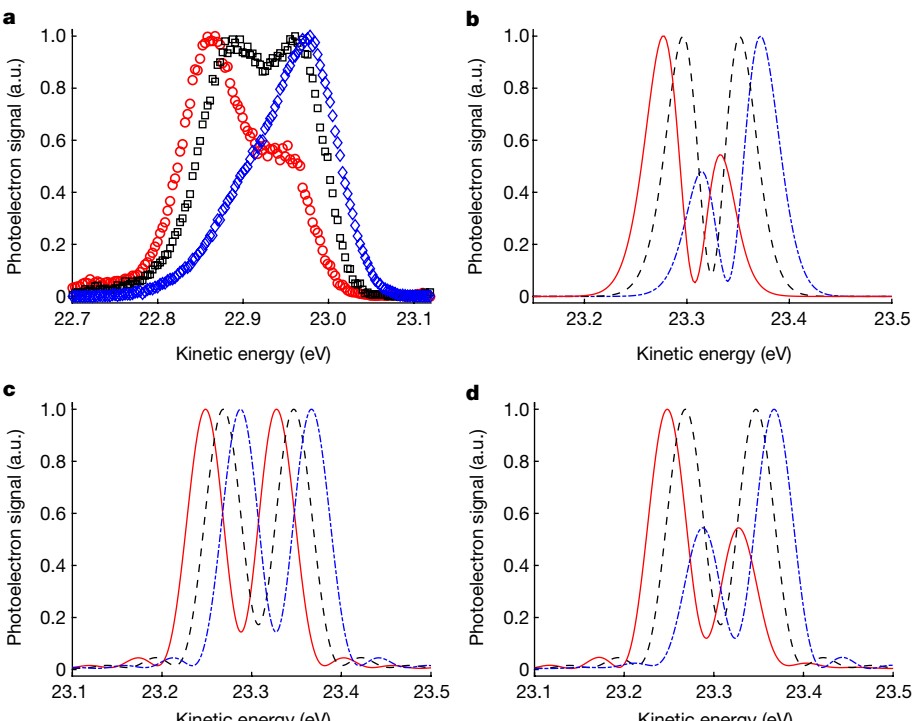

**Fig. 2 | Asymmetry of the ultrafast AT doublet. a**, Deconvoluted experimental photoelectron spectra with a symmetric AT doublet (black squares) at 23.753-eV photon energy, and the asymmetric ones at ±13-meV detuning (blue diamonds and red circles, respectively). **b**, Ab initio photoelectron spectra using TDCIS at three photon energies with a symmetric AT doublet at 24.157-eV photon energy (dashed black line) and the asymmetric ones at ±13-meV detuning. The red (blue) curve corresponds to red (blue) detuned light. **c,d**, The same as in **b**, but using the analytical model for 3/2 Rabi periods in the case of one-photon ionization from $|b\rangle$ (**c**) and two-photon ionization from $|a\rangle$ (**d**). The loss of contrast observed in the experimental spectra arises owing to macroscopic averaging of the target gas sample (see Supplementary Information for details). a.u., arbitrary units.

This blueshift is one of the major experimental results, and identifying its physical origin is among the main objectives of the theoretical efforts presented in this work. A strong asymmetry is observed when the FEL frequency is detuned to the red (red circles in Fig. 2a) or blue (blue diamonds in Fig. 2a) side of the symmetric doublet. The asymmetry of the AT doublet is qualitatively well reproduced by ab initio numerical simulations for helium within the time-dependent configuration-interaction singles (TDCIS) approximation[33], as shown in Fig. 2b. Gaussian pulses were used with parameters chosen to match the experimental conditions with an effective intensity of $2 \times 10^{13}$ W cm$^{-2}$ (as obtained from $\Omega$) and a pulse duration (FWHM) of 56 fs (see Methods for details). It is worth noting that the Rabi dynamics is sensitive to the exact shape of the driving pulse. For instance, a Gaussian pulse can induce more Rabi oscillations than a flat-top pulse with same FWHM by a factor of $\sqrt{\pi/(2 \ln 2)} \approx 1.5$ as follows from the area theorem[34]. Thus, the calculated photoelectron spectra from the analytical model using flat-top pulses in Fig. 1b for 3/2 Rabi periods agree well with those from the TDCIS calculations using Gaussian pulses with a FWHM close to a single Rabi period in Fig. 2b. Clearly, the AT doublet manifests itself between 1 and 3/2 Rabi periods. The difference in kinetic energy (about 0.4 eV) of the symmetric AT doublet between experiment and theory (Fig. 2a and Fig. 2b, respectively) is attributed to electron correlation effects not included in the TDCIS calculations that increase the binding energy beyond the Hartree–Fock level. The observed asymmetry in the AT doublet cannot be explained by a breakdown of the rotating-wave approximation because the experiment is performed at a resonant weak-coupling condition[6]: $\omega \approx \omega_{ba}$ and $\Omega/\omega_{ba} = 0.34\%$. Instead, we express the Rabi amplitudes from equation (1) in terms of their frequency components and find that $a(t)$ has two asymmetric components that are proportional to $(1 \pm \Delta\omega/W)$, whereas $b(t)$ has two symmetric components

that are proportional to $\pm\Omega/W$. Using the analytical model with 3/2 Rabi periods, we confirm that the AT doublet from $|b\rangle$ is symmetric, whereas that from $|a\rangle$ is asymmetric, as shown in Fig. 2c and Fig. 2d, respectively. Quite remarkably, the observed asymmetry in the experiment suggests that the photoelectron signal contains significant contributions from the two-photon ionization process from $|a\rangle$. We question how this is possible given that the electric field amplitude $E_0 = 0.02388$ atomic units implies an ionization-probability ratio of $10^4:1$ in favour of the one-photon process from $|b\rangle$.

We propose that the two-photon signal from $|a\rangle$ can compete with the one-photon signal from $|b\rangle$ owing to superposition of intermediate (complement) states, which are illustrated as grey bound and continuum states in Fig. 3a. This leads to a giant localized wave, $|\rho_{\neq b}\rangle$, compared with the normalized wavefunction for $|b\rangle$, as shown in Fig. 3b. The largest contributions to the giant wave come from the dipole-allowed complement states that are close to the one-photon excitation energy. The scaling factor owing to atomic effects is calculated to be about $1:10^4$ in favour of the two-photon process (see Extended Data Table 1 for the matrix elements). Thus, we can explain why the XUV-FEL pulse is intense enough for the non-resonant two-photon process from the ground state to be comparable to the one-photon process from the resonant excited state. In general, addition of two pathways leads to quantum interference that depends on their relative phase. From Fig. 3b, we notice that the giant wave oscillates out of phase with the excited state close to the atomic core, which affects the signs of the matrix elements (Extended Data Table 1). The ultrafast build-up of the AT doublet can be used to study the resulting interference phenomenon in time. To understand this phenomenon, we have used the analytical model to perform calculations where the one- and two-photon contributions are added coherently to simulate the angle-integrated measurements. In Fig. 3c, we show how the

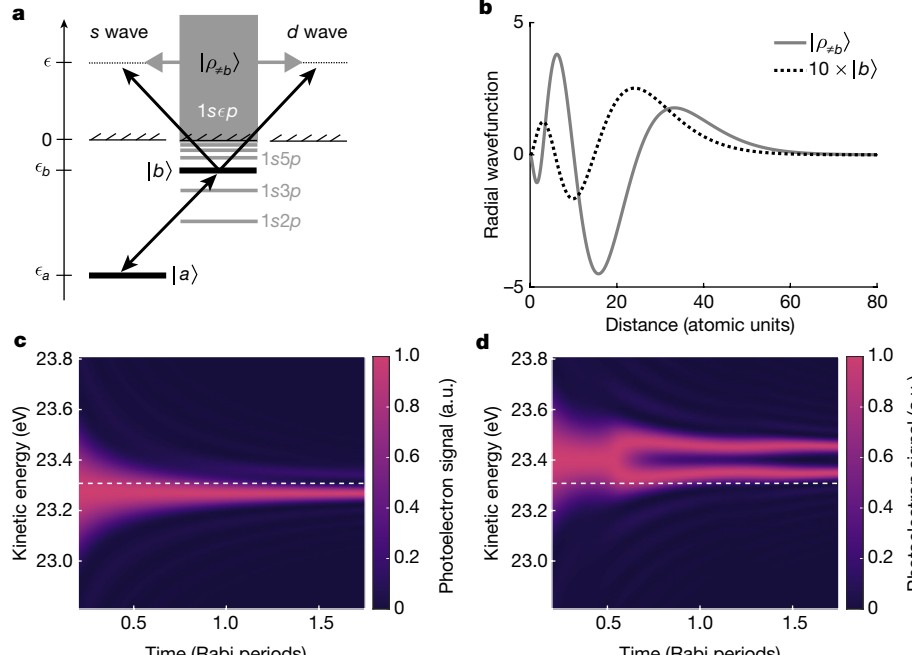

**Fig. 3 | Quantum interference with a giant wave. a**, Energy-level diagram for the photon transitions that lead to quantum interference. **b**, The summation of contributions from the non-resonant (grey) states in **a** leads to the formation of a giant wave, $|\rho_{\neq b}\rangle$. The excited state $|b\rangle$ is shown for comparison with a magnification factor of 10 (dotted black line). Both wavefunctions are computed for helium using CIS. **c**, Photoelectron spectra from the total analytical model containing contributions from both ground $|a\rangle$ and excited $|b\rangle$ states with resonant atomic excitation $\Delta\omega = 0$. **d**, The same as in **c**, but with $\Delta\omega = 62$ meV. The dashed white lines denote the expected kinetic energy (23.3076 eV) of a photoelectron that has absorbed two resonant photons. a.u., arbitrary units.

resonant case ($\Delta\omega = 0$) leads to a strongly asymmetric AT doublet after one Rabi period. Figure 3d indicates that a blue detuning ($\Delta\omega = 62$ meV) leads to the symmetric AT doublet at an earlier time between 0.5 and 1 Rabi periods. The advancement of the AT doublet in time follows from the faster Rabi cycling at the rate of generalized Rabi frequency. Thus, we have found that the blueshift of the symmetric AT doublet is due to quantum interference between the one-photon (I) and two-photon (II) processes. A general loss of contrast in the AT doublet structures is found by considering the effect of an extended gas target in our model. However, the two-photon doublet was found to be less sensitive to the volume averaging effect when compared with the one-photon doublet (see Supplementary Information for details), allowing us to clearly observe the AT doublet in the measured photoelectron signal.

To provide further evidence in support of the coherent interaction between the helium atoms and the XUV-FEL pulses, we show that the ultrafast emergence of the AT doublet can be interpreted in terms of the dressed-atom picture with coupled atom–field energies $\epsilon_\pm = (\epsilon_a + \epsilon_b + \hbar\omega \pm \hbar W)/2$. One photon energy above these coupled energies implies final photoelectron kinetic energies $\epsilon_\pm^{kin} = \epsilon_\pm + \hbar\omega$, where $I_p = -\epsilon_a = 24.5873$ eV is the ionization potential of helium. In Fig. 4a, kinetic energies are labelled with the uncoupled atom–field states[35], $|a, 1\rangle$ and $|b, 0\rangle$. The experimental results in Fig. 4b can be understood as one photon above $|a, 1\rangle$ at large detuning of the XUV-FEL pulse. This is because the interaction is weak far from the resonance with the atom remaining mostly in its ground state $|a\rangle$, such that two photons are required for photoionization. The region closer to the resonance is influenced by quantum interference between the one-photon (I) and two-photon (II) processes that leads to suppression of $|b, 0\rangle$ and enhancement of $|a, 1\rangle$, with both coupled energies appearing briefly to form an avoided crossing in kinetic energy. It is noted that the avoided crossing appears at a blue detuning from the resonant transition, $\Delta\omega = 0$ (denoted by the dashed vertical line), revealing the quantum interference between the two

pathways from the ground state $|a\rangle$ and the excited state $|b\rangle$. Similar results were obtained from the TDCIS simulations (Fig. 4c) and the analytical model with contributions from both $|a\rangle$ and $|b\rangle$ for 3/2 Rabi periods (Fig. 4d). The observed blue detuning for the experimental avoided crossing (about 11 meV) is well reproduced by TDCIS calculations (about 14 meV). The enhanced shift of the AT doublet to blue detuning in the analytical model is an effect of the pulse envelope that can be reproduced with TDCIS using smoothed flat-top pulses. This indicates that the amount of blueshift of the AT doublet can be coherently controlled by the exact profile of the FEL pulse.

Our results show that it is now possible to simultaneously drive and interrogate ultrafast coherent processes using XUV-FEL pulses. Previous attempts to understand Rabi dynamics at short wavelengths have relied on the strong-field approximation, where the influence of the atomic potential is neglected, leading to an inconsistent AT doublet when compared with numerical simulations[19]. In contrast, our analytical model includes the full effect of the atomic potential and Rabi dynamics in the two-level subspace, whereas the remaining transitions to and within the complement of the Hilbert space are treated by time-dependent perturbation theory. Consequently, we could establish a unique mechanism in the form of a giant Coulomb-induced wave from the ground state to explain why the non-resonant two-photon process can compete with the resonant one-photon process and generate quantum interference effects at the high intensities provided by the XUV-FEL beam. With this model, we now understand how ultrafast Rabi dynamics at short wavelengths are imprinted on photoelectrons from weakly ionized atoms. Our experimental approach of using photoionization as an in situ probe of Rabi dynamics does not rely on any additional laser probe field, and, hence, is easily applicable to other quantum systems. We think that such photoionization processes are of interest to different domains: the one-photon domain connected to the symmetric excited-state dynamics (I) and the two-photon domain connected to the asymmetric ground-state dynamics (II), as shown in

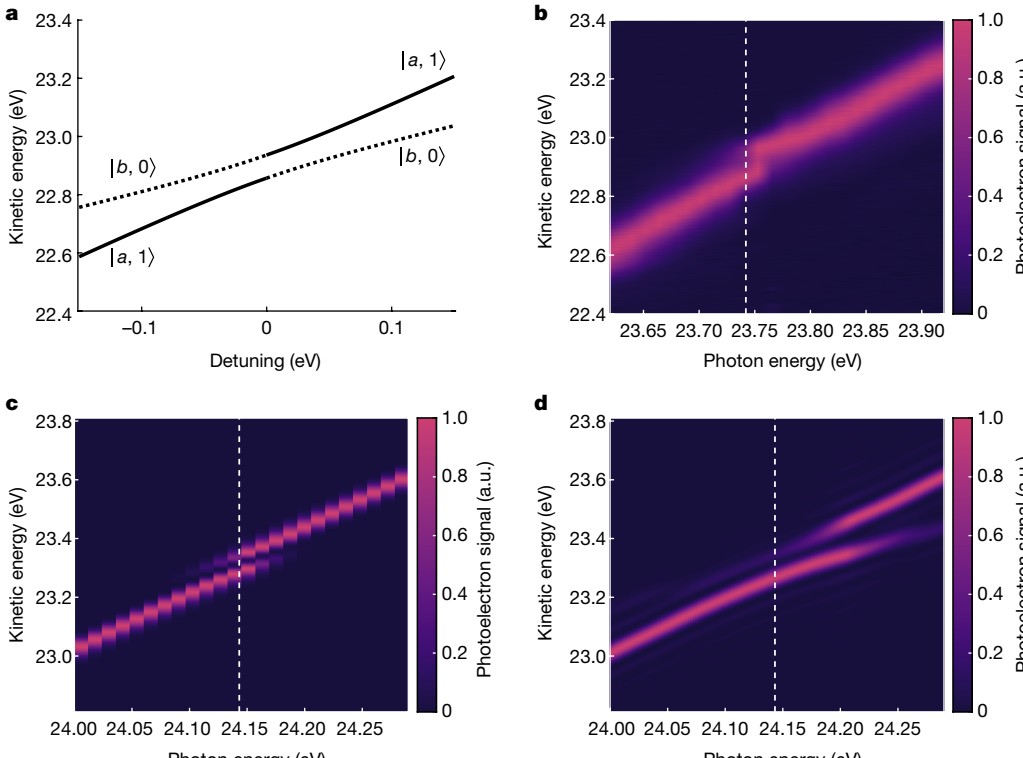

**Fig. 4 | Avoided crossing phenomena in the energy domain. a**, Photoelectron kinetic energies, for one photon above the energy of the dressed-atom states, as a function of detuning. **b**–**d**, Photoelectron spectra as a function of the photon energy retrieved experimentally (**b**), using TDCIS (**c**) and using the total analytical model for 3/2 Rabi periods (**d**). In each case, the dashed white line

corresponds to the photon energy for the $1s^2 \rightarrow 1s4p$ transition in helium. The shifts between the energy scales of **a** and **b**, and the energy scales of **c** and **d**, are due to the difference between the experimental and the Hartree–Fock ionization potential. a.u., arbitrary units.

Fig. 1c. At the boundary between these domains, quantum interference is manifested in the photoelectron signal. The experimental blueshift of the symmetric AT doublet, reported in our work, is an observation of this type of effect. Owing to substantially different angular distributions from domains I and II (Extended Data Table 1), we predict that the boundary regime will exhibit intricate angular dependencies of the photoelectrons that could be the subject of upcoming experiments. Studying more complicated Rabi dynamics, affected by rapid photoionization[36] or autoionization decay[37], are natural extensions of our work. Coherent population inversion with core-excited states and ultrafast core ionization with simultaneous ionic-state excitation[21] are examples of configurations achievable at short wavelengths. Given the ongoing developments of seeded FEL facilities around the world[38,39] capable of providing light pulses down to few-ångström wavelength, our findings can inspire future studies involving coherent control in multi-electronic targets, such as molecules, and nano-objects with site specificity, opening up pathways for steering the outcomes of photo-induced processes across ultrafast timescales.

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

## Methods

### Experiment

The experiment was carried out at the low-density-matter beamline of FERMI[40]. A pulsed Even–Lavie valve, synchronized with the arrival of the FEL pulse served as the target source. The target gas jet was estimated to be a cone with a diameter of 2 mm at the interaction region. We measured the photoelectron spectra at and around the $1s^2 \to 1s4p$ transition in helium, using a 2-m-long magnetic bottle electron spectrometer (MBES). The gas jet, FEL beam and magnetic bottle axes are mutually perpendicular, with the first two being on the horizontal plane of the laboratory, and the last one in the vertical direction. Before entering into the flight tube of the MBES, the photoelectrons were strongly retarded to below 1-eV kinetic energy to achieve high spectral resolution ($E/\Delta E \approx 50$). To suppress any short-term fluctuation arising from the instability of the FEL, we performed a 'round trip' scan across the wavelength range, 52.50 nm ↔ 51.80 nm. Empirically, the FWHM of the XUV-FEL pulse duration ($\tau_{xuv}$) can be approximated[32] to be in between ($\tau_{seed}/n^{1/2}$) and ($7\tau_{seed}/6n^{1/3}$). Here $\tau_{seed} \approx 100$ fs is the duration (FWHM) of the seed pulse (wavelength 261.08 nm) and $n = 5$ is the harmonic order for the undulator. It leads to $\tau_{xuv} = 56 \pm 13$ fs, which matches well the FWHM of about 66 fs, obtained from the simulation of the FEL dynamics using PERSEO[41]. The spectral bandwidth (FWHM) of the pulse was estimated using PERSEO to be around 0.13 nm at the central wavelength of $\lambda = 52.216$ nm. Extended Data Fig. 1a and Extended Data Fig. 1b show the simulated spectral and temporal profiles of the FEL pulse, respectively. At best focus, the spot size (FWHM) was estimated to be 12 μm. We measured the energy per pulse at the output of the FEL undulator to be around 87 μJ, which refers to the full beam including all photons contained in the transverse Gaussian distribution. To consider those, we used $4\sigma$ as the FEL beam diameter at best focus, where $\sigma = 12/2.355 \approx 5.1$ μm. Hence, the beam waist ($w_0$) is given by $w_0 = 2\sigma = 10.2$ μm, along with a Rayleigh length of $\pi w_0^2/\lambda \approx 6.3$ mm. The peak intensity was estimated to be about $1.4 \times 10^{14}$ W cm$^{-2}$. This clearly shows that the FEL pulses were intense enough to drive the coherent Rabi dynamics. However, this peak intensity alone does not directly correlate to the ultrafast Rabi dynamics, as the Rabi frequency is proportional to the **E**-field strength, and Rabi cycling requires sufficient area of the pulse for the AT doublet to emerge. Furthermore, given the shot-to-shot fluctuations of the FEL pulse parameters, we have used the observed AT splitting to extract the average interaction strength in a reliable manner without any need for simulated values of the pulse duration or experimental mean pulse energy.

### Data analysis

To filter the measured photoelectron spectra on a shot-to-shot basis, we used the photon spectrum recorded by the Photon Analysis Delivery and REduction System (PADRES) at FERMI to determine the bandwidth (FWHM) of the XUV pulse. Any shot without the photon spectrum was rejected: out of 355,000 shots, 354,328 shots were retained. All the shots with more than 65-meV FWHM width were discarded (Extended Data Fig. 2a). It is noted that the simulated value of the photon bandwidth (59 meV) lies within the filtering window of 20–65 meV. In addition, we chose only the shots with integrated spectral intensities ranging from $0.8 \times 10^5$ to $1.6 \times 10^5$ in arbitrary units (Extended Data Fig. 2b). The filtered shots were sorted into 30 photon-energy bins, uniformly separated from each other by about 13 meV and covering the entire photon energy window of the wavelength scan (Extended Data Fig. 2c). Overall, only 304,192 shots (filtering ratio of 0.857) out of the raw data were retained. The measured photoelectron spectra, following shot-to-shot filtering, are shown in Extended Data Fig. 3. The avoided crossing is only faintly visible here. To obtain the clear avoided crossing from Fig. 4b, we deconvoluted the photoelectron spectra for three photon energies near the $1s^2 \to 1s4p$ transition using the Richardson–Lucy blind iterative algorithm[42]. To reduce the noise introduced during the deconvolution, we incorporated the Tikhonov–Miller regularization procedure into the algorithm[43]. The outcomes are shown in Extended Data Fig. 4. Following deconvolution, the values of FWHM for the Gaussian instrument-response functions were found to be 70.9 ± 1.2 meV, 69.6 ± 2.4 meV and 69.4 ± 1.4 meV, for the three photon energies. These values match well the combined resolution of about 65 meV, obtained from the photon bandwidth and the kinetic energy resolution of the MBES. No filter, either metallic or gaseous, was used along the path of the FEL beam. Hence, a minor contribution (<5%) from the second-order light can be noticed as an asymmetric tail close to 22.8-eV kinetic energy (Extended Data Fig. 4a,b). To rule out any artefact from the fluctuations of the FEL pulse properties, we used another filtering criterion for the photon bandwidth (1–45 meV) and the integrated spectral intensity ($1 \times 10^5$ to $3 \times 10^5$ a.u.). The corresponding deconvoluted photoelectron spectra at 23.753 eV is shown in Extended Data Fig. 5, along with that from Fig. 2a. No significant change in the AT doublet structure due to change in filtering criteria could be seen. Finally, for a transform-limited Gaussian pulse, $\tau_{xuv}$ can vary between 30 fs and 90 fs from shot to shot that encompasses its empirical value of 56 ± 13 fs. As $\tau_{xuv}^2$ is significantly higher than the absolute value of the simulated group-delay dispersion of the FEL pulse of −690 fs$^2$, no effect due to the linear chirp was considered in the theoretical calculations.

### Numerical simulations using TDCIS

The ab initio numerical simulations are performed using the time-dependent (TD) configuration-interaction singles (CIS) method[33,44–46] in the velocity gauge. The CIS basis for helium is constructed using Hartree–Fock orbitals that are computed using B-splines. Exterior complex scaling is used to dampen spurious reflections during time propagation of TDCIS[47]. The vector potential of the XUV-FEL pulse is defined as

$$A(t) = A_0 \sin(\omega t)\exp\left[-2\ln(2)\frac{t^2}{\tau^2}\right]. \qquad (2)$$

The central frequency, $\omega$, is set close to the CIS atomic transition frequency, $\omega_{ba} = 0.887246$ atomic units = 24.1432 eV, between the Hartree–Fock ground state $|a\rangle = 1s^2$ ($^1S_0$) and the singly excited state $|b\rangle = 1s4p$ ($^1P_1$). The duration of the pulse is set to $\tau = 56$ fs and the peak intensity is set to $I = (\omega A_0)^2$ [a.u.] $\times 3.51 \times 10^{16}$ [W cm$^{-2}$] $= 2 \times 10^{13}$ W cm$^{-2}$. The CIS dipole matrix element between the ground state and the excited state is given by $z_{ba} = 0.124a_0$ and the ionization potential is related to the $1s$ orbital energy in Hartree–Fock $I_p = -\epsilon_a = 24.9788$ eV, in accordance with Koopmans' theorem. Photoelectron distributions are captured using the time-dependent surface flux (t-SURFF)[48] and infinite-time surface flux (iSURF)[49] methods. The high kinetic energy of the photoelectrons ensures a proper description of the physics by the surface methods.

### Data availability

Raw data were generated at the FERMI large-scale facility. Derived data supporting the findings of this study are available from the corresponding authors upon request.

### Code availability

Codes used in this study are available from the corresponding authors upon reasonable request.

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

**Acknowledgements** We acknowledge financial support from LASERLAB-EUROPE (grant agreement number 654148, European Union's Horizon 2020 research and innovation programme). S.N. thanks CNRS and Fédération de Recherche André Marie Ampère, Lyon for financial support. J.M.D. acknowledges support from the Knut and Alice Wallenberg Foundation (2017.0104 and 2019.0154), the Swedish Research Council (2018-03845) and the Olle Engkvist's Foundation (194-0734). R.F. thanks the Swedish Research Council (2018-03731) and the Knut and Alice Wallenberg Foundation (2017.0104) for financial support. P.E.-J. acknowledges support from the Swedish Research Council (2017-04106) and the Swedish Foundation for Strategic Research (FFL12-0101).

**Author contributions** E.O. and M.B. contributed equally to this work. D.B., C.C., M.D.F., P.E.-J., R.F., G.G., M.G., S.M., L.N., J.P., O.P., K.C.P., R.J.S., S.Z. and S.N. performed the experiment and collected the data. S.N. analysed the data. M.D.F., O.P. and C.C. managed the low-density-matter end-station. R.J.S. and R.F. operated the magnetic bottle electron spectrometer. L.B., M.B.D., F.S. and L.G. optimized the machine. M. Manfredda and M.Z. characterized the pulses. P.V.D. contributed to the analysis of obtained results. M. Meyer and C.M. contributed to the initial planning of the project. E.O., M.B., S.C., F.Z. and J.M.D. provided the theoretical calculations. E.O. developed the analytical model and M.B. carried out the TDCIS calculations, under the supervision of J.M.D. S.N. and J.M.D. wrote the manuscript, which all authors discussed. S.N. proposed and led the project.

**Funding** Open access funding provided by Lund University.

**Competing interests** The authors declare no competing interests.

**Additional information**
**Correspondence and requests for materials** should be addressed to Saikat Nandi or Jan Marcus Dahlström.

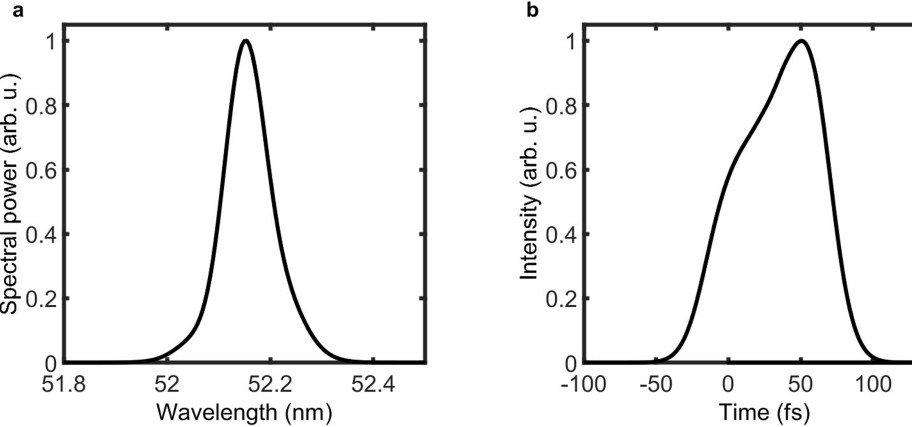

**Extended Data Fig. 1 | Simulated FEL pulse properties. a,b,** Spectral (**a**) and temporal (**b**) profiles of the XUV-FEL pulse as obtained from the simulations using PERSEO.

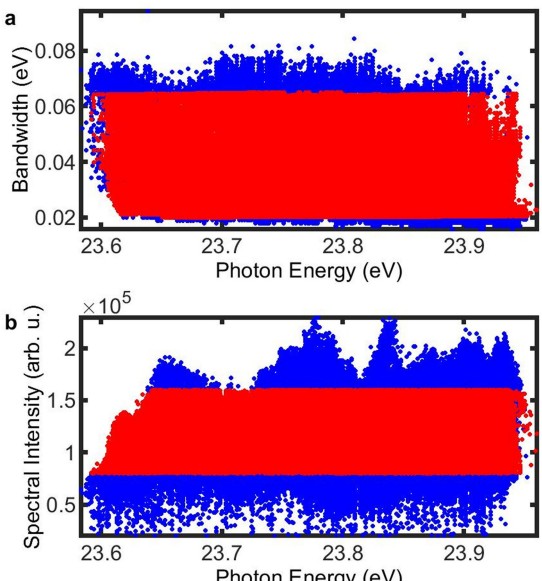

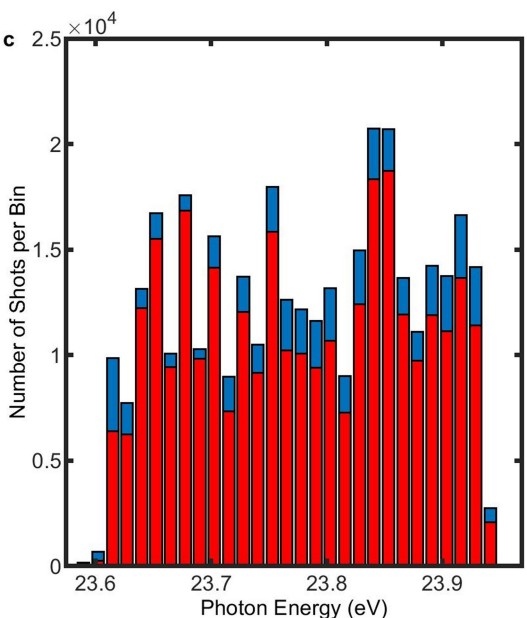

**Extended Data Fig. 2 | Filtering criteria for the measured data. a**, Shot-to-shot variation of the FEL bandwidth (FWHM) as a function of the photon energy. The blue dots represent the measured FWHM using the PADRES spectrometer and the red dots represent the filtered shots. **b**, Same as **a**, but for the integrated spectral intensity. **c**, All the shots are distributed over 30 equally spaced (spacing: ~13 meV) photon-energy bins, spanning the entire range of wavelength scan. The red ones correspond to the filtered shots satisfying both the criteria in panel **a**–**b**.

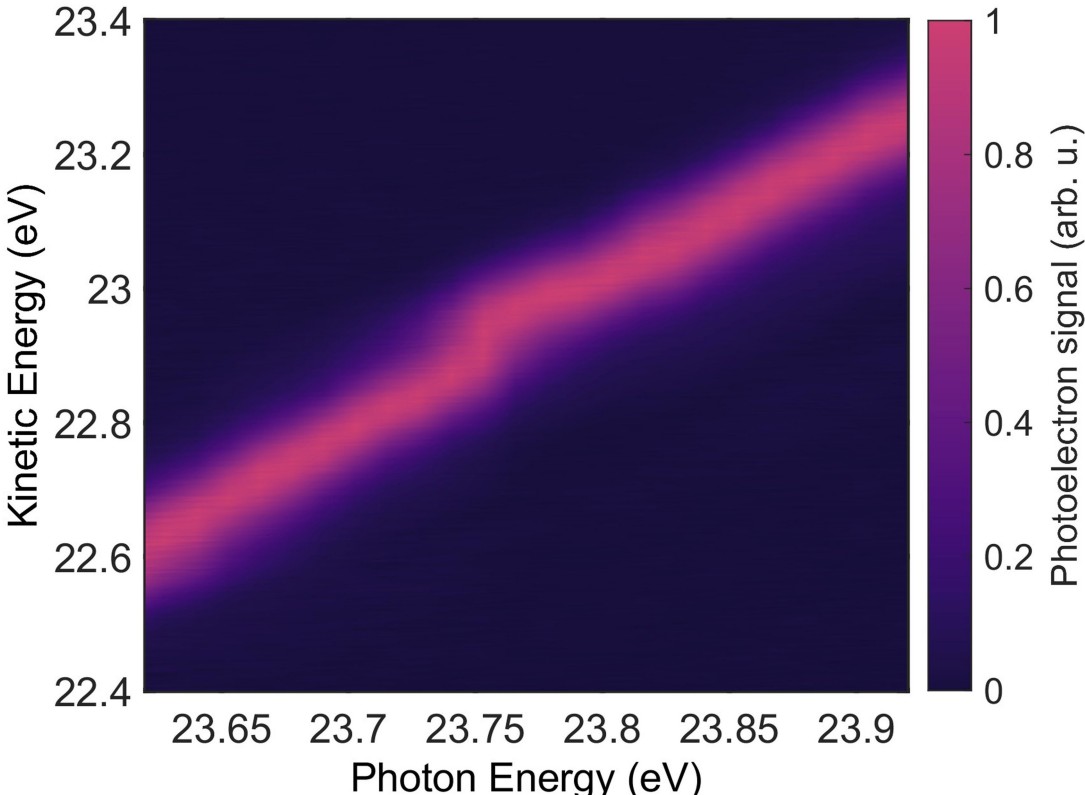

**Extended Data Fig. 3 | Avoided crossing without deconvolution.** Measured photoelectron spectra, as a function of the photon energy, without any deconvolution procedure performed. Notice the faint avoided crossing.

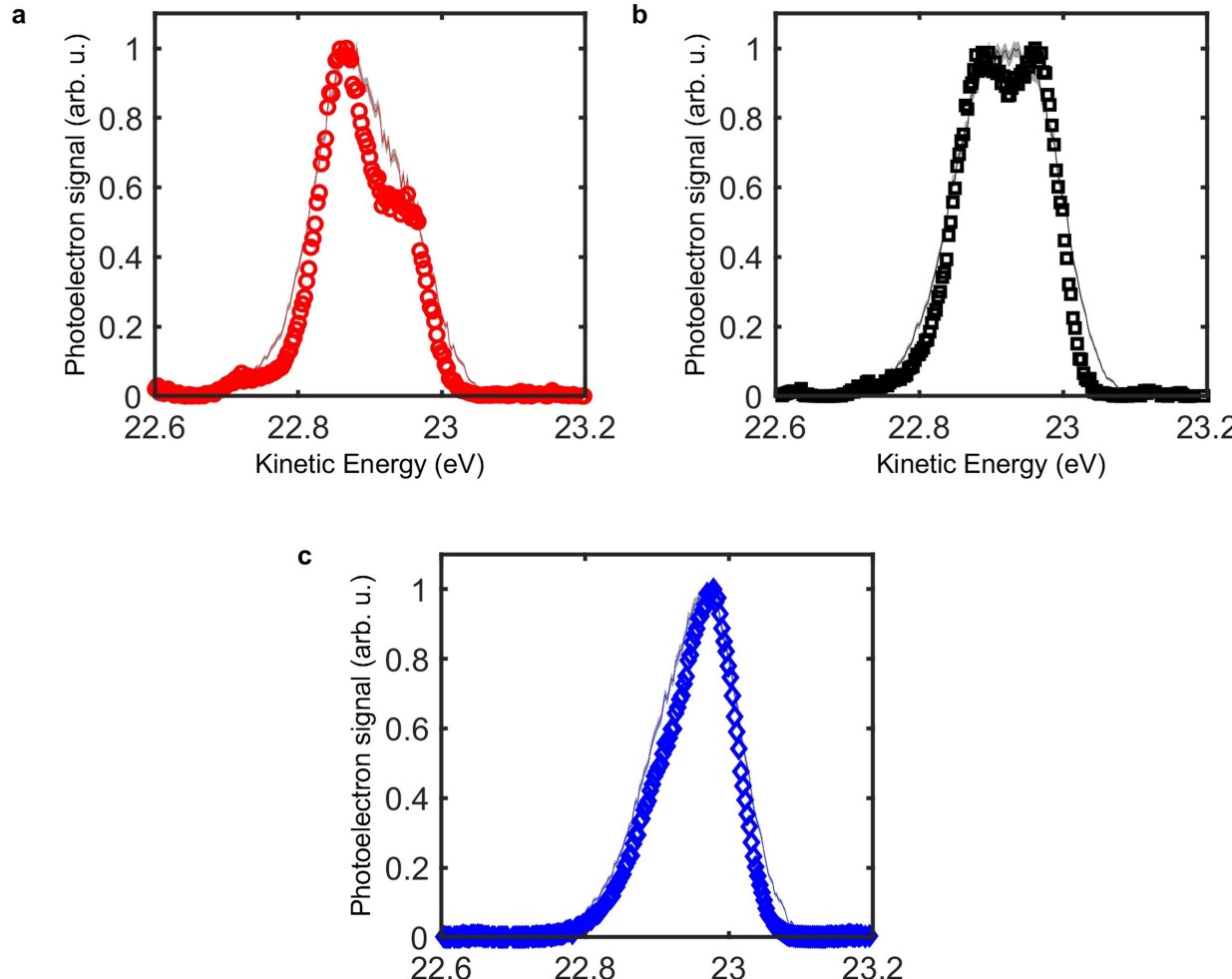

**Extended Data Fig. 4 | Experimental photoelectron spectra with and without deconvolution. a**, Experimental photoelectron spectra at 23.740 eV photon energy. Solid line: raw data and circles: deconvoluted form. **b**, Same as **a**, but at 23.753 eV photon energy. Here, the solid line represents the raw data and the squares represent the deconvoluted spectrum. **c**, Same as **a**–**b**, but at 23.766 eV photon energy, where once again the solid line constitutes the raw data and the diamonds are for its deconvoluted form. In each case, the open symbols are same as shown in Fig. 2a of the main text. The shaded region in each sub-panel represents the corresponding Poisson fluctuations of the photoelectron signal.

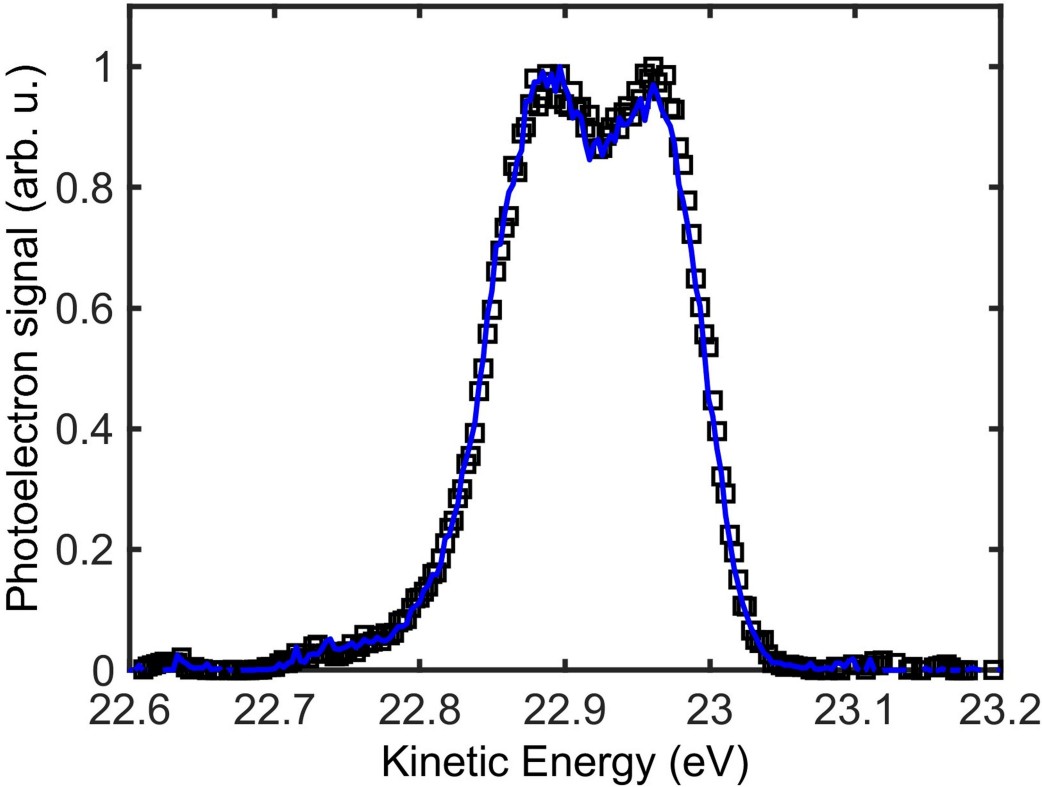

**Extended Data Fig. 5 | Deconvoluted spectra with two different filtering criteria.** Deconvoluted experimental photoelectron spectra at 23.753 eV for two different filtering criteria. Open black squares: for photon bandwidth of 20–65 meV and integrated spectral intensity of $0.8 \times 10^5 – 1.6 \times 10^5$ (same as in the Fig. 2a of main text). Solid blue line: for photon bandwidth of 1–45 meV and integrated spectral intensity of $1 \times 10^5 – 3 \times 10^5$.

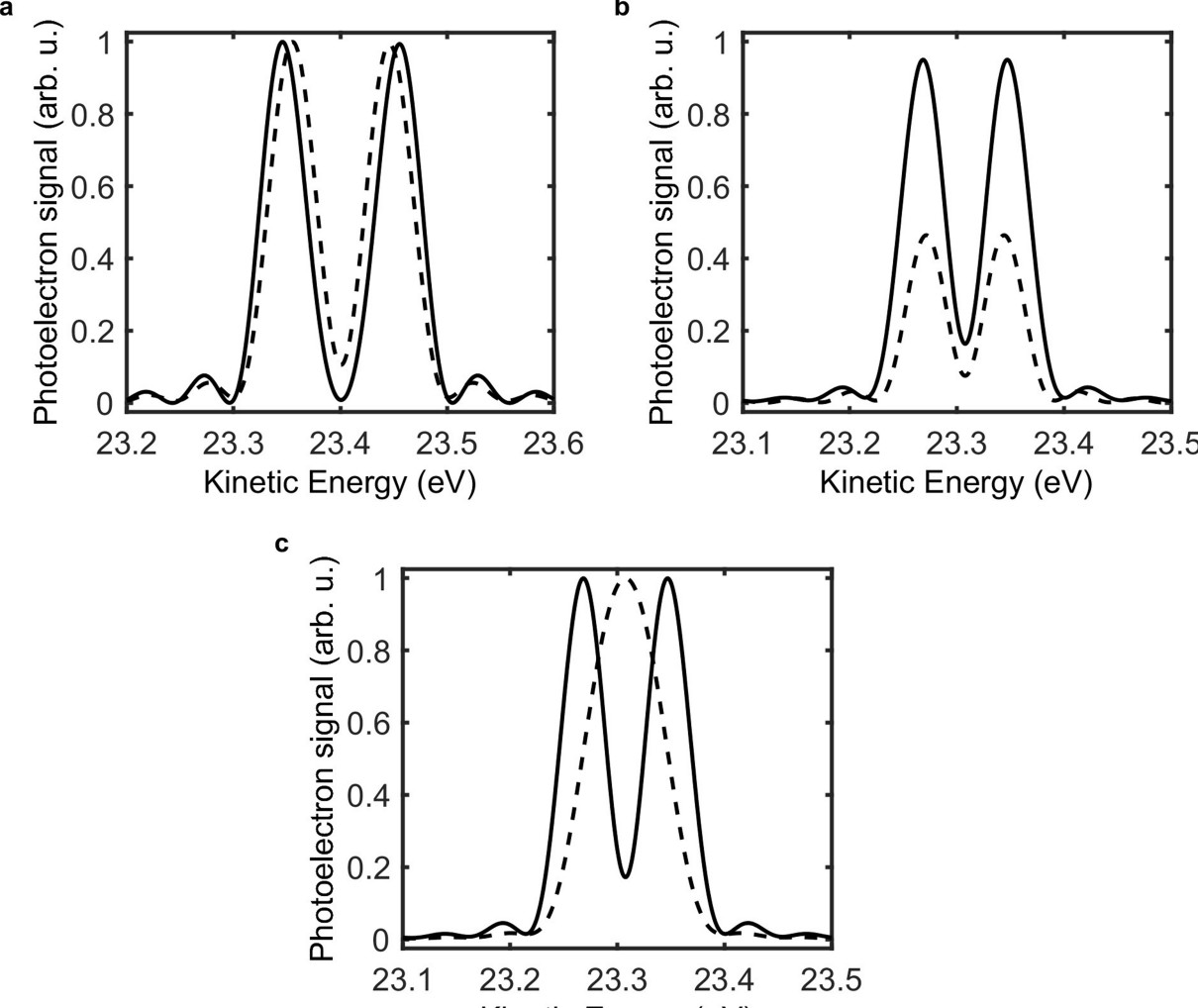

**Extended Data Fig. 6 | Effects of intensity averaging on the photoelectron spectrum. a**, Photoelectron spectra generated with the analytic model for a single atom (solid line), and for a macroscopic sample (dashed line). A pulse length of 3/2 Rabi periods, and detuning of $\Delta\omega = 62$ meV was used. **b-c**, Same as **a**, but for the individual contributions of the two- and one-photon processes, respectively. The results in both **b**, and **c** are calculated for $\Delta\omega = 0$ meV, where the spectra are symmetric. The separate contributions are normalized to the maximum of the one-photon spectra for both the single-atom and intensity averaged signals.

**Extended Data Table 1 | Dipole transition elements**

| Final state | $\lvert b \rangle$ | $\lvert \rho_{\neq b} \rangle$ |
|:---:|:---:|:---:|
| $s$-wave | 0.009311 | 0.1056 |
| $d$-wave | 0.01298 | -1.300 |

Values of the dipole transition elements $z_i^f$ computed using CIS functions.