## [Peer Review File · Nature]

Manuscript Title: Observation of Rabi dynamics with a short-wavelength free-electron laser

Reviewer Comments & Author Rebuttals

Reviewer Reports on the Initial Version:

Referees' comments:

Referee #1 (Remarks to the Author):

In their manuscript, the authors describe an experiment on the dynamics of helium under the action of a seeded free electron laser in the XUV frequency regime. The focus is on Rabi oscillations, i.e., coherent periodic modulations of the occupation of two quantum states. The authors state that so far, such oscillations induced by short wavelength radiation have not yet been directly observed. Their main claim is that they induce such oscillations using XUV pulses, and observe them via the photoemission of electrons.

First of all, I am not convinced about the significance and novelty of the results. The introductory paragraph suggest the lack of a direct observation of Rabi oscillations at short wavelengths as the open challenge to be addressed. However, I do not think that the present experiment fills this gap. In my view it is not justified to speak of a direct observation because the entire analysis relies on heavy theory input. Without the comparison of complicated signatures to an extended theory model, it is impossible to pinpoint the existence of oscillations. The agreement between theory and experiment, e.g., in Fig 2, is only qualitative. The avoided crossing (Fig 4) is (seemingly) simpler to interpret, but it does not show any appreciable population transfer.

However, the reported results do go beyond what has been reported before in that the claimed oscillations are induced by the XUV field. Previous experiments also involved XUV fields, but employed lower-frequency (IR) fields to induce the Rabi oscillations. This is indeed a difference, but it is not clear to me in how far this opens up new avenues - the manuscript does not really comment on this question. I also fail to see how the results generate new understanding of the XUV-induced dynamics. The basic condition for Rabi oscillations is to have coherence- and population losses low enough such that the coherent dynamics is faster than the incoherent one. In this regard, there is no difference between the original Rabi model and the present paper. The challenge at higher photon energies only is that the typically required Rabi frequencies can not be realized without simultaneously inducing strong loss processes, such that a delicate balance is required. But this, of course, is also long known.

I am even tempted to challenge whether "Rabi oscillations" have been observed at all. There are several reasons for this:

- The observation of oscillations requires that the coherence between ground and excited state is maintained throughout the dynamics. The authors argue that this is possible since the life time of

the excited state is 1ns, much longer than the time scale of the experiment governed by the 50fs pulses. But this comparison is misleading, in my view. First of all, obviously the 1ns lifetime only is valid in the absence of external fields. In the presence of the fields, the photo ionization both, of the ground and excited state, like reduces the lifetime of the coherence also to the ~ 10 fs scale or even below. In this situation, it is not clear if the oscillations persist, or if only a "damped" transfer from the ground to the excited state takes place. The authors do not comment on this issue. They also do not comment on the challenge that it may not be possible to go to higher Rabi frequencies (i.e., more oscillations) due to the inevitable related increasing ionization, but instead praise their main source of decoherence and population loss (ionization) as an in-situ probe. Second, at least in the traditional Rabi model, the lifetime of the upper state is not really an issue by itself. Rather, the ratio of the Rabi frequency to the loss rates should be the key quantity which determines whether oscillations appear or not? But the authors do not comment on the Rabi frequency in their qualitative discussion on page 4.

- Furthermore, the archetype two-level system of the Rabi model is strongly modified in the present experiment by other states coupled by the applied field - see Fig. 3(a). The interpretation of the photoelectron spectra in Fig 2 is only possible if this extended level scheme is considered. The authors stress the significance of other excited states (with the "giant wave" which is not really explained in the manuscript in a way that would make it accessible to a more general reader, let alone its effect on the dynamics in the a/b subspace). Again, it is not clear if there are really oscillations between $|a\rangle$ and $|b\rangle$.

- Only after having discussed Rabi oscillations in detail, the authors state on page 5 that their experiment is close to a single Rabi cycle. I am not sure if the majority of readers would associate this to "Rabi oscillations" promised in the introduction. Note that the short pulse duration does not restrict the number of Rabi cycles a priori because in principle one could resort to higher Rabi frequencies than what has been achieved in the manuscript.

* The photoelectron kinetic energies in Fig. 4 even suggest that there is no appreciable excitation of the upper state $|b\rangle$ at all. There is no second resonance associated to the dressed state asymptotically belonging to $|b,0\rangle$. The authors write "This is because the interaction there is weak and the atom remains mostly in its ground state". This again raises the question whether oscillations really have been observed. Further, is it really justified to speak about dressed states in this situation? If one diagonalizes the 2-level Hamiltonian, then both eigenstates should contain significant population even in the limit of low excitation?

- The manuscript does not relate to traditional key signatures of Rabi oscillations, or experimental approaches to verify them. For example, a typical approach would be to vary the field intensity, while keeping the field on resonance. This way, the change of the oscillation period as function of time could be observed. Of course, time-resolved measurements are challenging with fs pulses. But a variation of the pulse intensity would be feasible, and from the interpretation given by the authors in Fig 1, it should allow one to observe qualitatively different photoelectron spectra as function of the pulse area. Instead, the experiment varies the detuning of the field. This also does change the *generalized* Rabi frequency $\sqrt{\Delta^2 + \Omega^2}$ and thus the oscillation frequency. But at the

same time, it also modifies the oscillation amplitude, even in the ideal case, and potentially also relative phases between the different ionization channels are modified on the $1/\text{detuning}$ time scale, which are not accounted for in the interpretation in Fig 1. As a result, the detection is modified as well. I do not think that the present experimental results do allow one to disentangle the effects of the actual desired dynamics (Rabi oscillations) from those of the modified detection. For example, what is the meaning of an atom performing $3/2$ Rabi oscillations if it continuously undergoes ionization on similar time scales? Or, would it be possible that the line shape changes in Fig 2 are not due to Rabi oscillations /AT splitting, but due to varying phases induced by the detuning of different ionization channels, like in Fano resonances? (One should also note that the energy of the symmetric configuration differs between experiment and theory.)

- The experiment is in an entirely different regime than traditional Rabi flopping experiments, due to the highly transient nature. This is not necessarily a disadvantage, but one may wonder in how far a direct relation to the original Rabi model is justified. For example, it is reported that the Rabi dynamics in the present setup is sensitive to the exact shape of the driving pulse. On the contrary, in the resonantly driven original Rabi model, the dynamics only depends on the pulse area (area theorem), and not on the pulse shape. Further, the Rabi frequency - and with it the AT splitting - is highly transient in the present experiment. Why would one still expect a splitting as shown in Fig 2 or in Fig 4, even though the ionizing field is always present in the experiment (unlike in previous work which relied on a time-delayed ionizing pulse)? Also, the AT splitting should be time-dependent as well, ranging from 0 to the maximum value.

On a more technical side, the shape of the experimental and the theoretical curves in Fig 2 are somewhat different. In particular, there are pronounced "dips" in the theoretical curves which are not present in the experimental data. What is the reason for this? If the dips are due to interferences, then the missing dips could be interpreted as arising from (partially) incoherent dynamics.

Since the analysis relies on heavy theory anyway, I wonder whether it would be possible to calculate the actual dynamics projected into the $|a\rangle$, $|b\rangle$ subspace for the experimentally relevant parameters. The normalization should be such that 100% refers to the initial population in the ground state. This way, the population losses out of the two-level system, and the presence of oscillations (or damped dynamics) could be judged. This may not be completely reliable since the theory predictions differ somewhat from the experimental data, but if no clear oscillations would be visible in the theory, then one should safely be able to reject the notion of population oscillations in the experiment.

Regarding the experiment, is there sufficient data at different pulse intensities but fixed detuning? Would this allow for an analysis of the dependence of the Rabi oscillations on the pulse area/peak intensity? This could form a more direct signature of an oscillation.

Overall, I cannot recommend publication of the manuscript in Nature for the above reasons.

Referee #2 (Remarks to the Author):

The authors present photo-electron spectra of Helium driven with high-intensity XUV radiation close to the 1s to 4p resonance. The intensity of the XUV pulses is high enough to induce one to two Rabi Oscillations between the ground and excited state during the duration of the pulse. The post-processed spectra show a splitting of the photo emission line, reminiscent of Autler-Townes doublets. Moreover, the evolution of the spectrum as a function of detuning shows a feature hinting towards an avoided crossing. The interpretation of the experimental relies on a substantial theory effort: Time-dependent Configuration Interaction Singles (TDCIS) calculations qualitatively reproduce the features of the experiment. Moreover, an analytical model based on time-dependent second-order perturbation theory on top of the two-level Rabi dynamics is presented and serves as the basis of the interpretation of the experimental photoelectron data. The measured line shapes are explained as a dynamical interference effect between a 1-photon ionization pathway from the excited state and a 2-photon ionization from the ground state. Interestingly, the authors identified an effective intermediate state in the 2-photon transition featuring a giant effective dipole moment that leverages the generally weak 2-photon process. The arguments and supporting theory fully corroborate the interpretation of the line shapes in terms of Rabi Oscillations. All the appropriate credit has been given and previous work was cited accordingly. This work demonstrates a wonderful interplay between theory and experiment.

Originality and significance: The authors could, for the first time, demonstrate Rabi Oscillations in EUV bound-to-bound transitions, by studying their manifestation in photoelectron spectra. These kind of experiments are extremely difficult to achieve at short-wavelength free-electron laser sources. By using the seeded FERMI FEL, previous limitations to show the effect of Rabi oscillations at SASE FELs of intrinsically limited temporal coherence could be mitigated. The discovered dynamical interference pathway of 1- and 2-photon transitions of the driven atomic systems could in the future eventually, if set-up in pump-probe schemes, develop into powerful phase-sensitive tool to study electron-wave packet dynamics. It remains questionable, if the effect of the “giant intermediate wave”, that is necessary to observe strong coherence effects, is a general feature, or particular to Helium.

Data & methodology: The authors did a careful data analysis, transparently presented in the manuscript. Moreover, two distinct theory results have been presented – a time-dependent configuration interaction calculation and an analytical model, the basic building block on which the interpretation of the experimental findings rests.

Appropriate use of statistics and treatment of uncertainties:

- In order to get an idea about the standard deviation of the experimental data, it would recommend the authors to include according bars in Extended Data Figure 4. Does the deconvolution algorithm allow to also assign error bars to the processed data?
- Figure 3 c and d: A false-color scale should be added. Is the color intensity shown in the figures according to a linear or logarithmic scale? A logarithmic scale could maybe enhance the visibility of the additional Autler-Townes peaks that are visible in d).
- Figure 4 b-d: I suggest the same improvements as for 3 c and d

Suggested minor improvements:

- At several points in the main text (page 5 line 73 and page 7 line 118, page 8 line 140,) the authors refer to the “required” blue shift of the applied frequency in order to produce symmetric photoelectron lineshapes/position of avoided crossing. Why mentioning this fact twice? What can be learned from this shift? Why is it important? Can one experimentally determine a relative value of the effective transition matrix elements/cross sections of one- versus two-photon ionisation?
- The title “Studying ultrafast Rabi dynamics with short-wavelength seeded free-electron laser” might be a bit misleading, since no time-dependent measurements have been made. I would suggest to replace “dynamics” with “oscillations” in the title.
- In the abstract (introductory paragraph) the authors refer to “The measured photoemission signal revealed...” Since it could be ion- or electron signal, I would suggest to be more specific and write “The measured photo-electron spectra revealed...” or similar referring to the photoelectron.
- In the introductory paragraph, instead of stating “using theoretical analyses that go beyond the strong-field approximation...” I would rather refer to the actual theory that has been developed, i.e. time-dependent perturbation theory on top of the Rabi dynamics, or similar.
- Line 77, page 5: I suppose that “quantitatively” should actually read “qualitatively”.
- Line 114, page 7: “to simulate angle-integrated measurements” The mentioning of an angular integral here is somehow ad hoc and there should not be much angular dependence, starting out with a spherically symmetric He groundstate. Why referring to the angular integral here? Could it be that the average over the spatial interaction volume is meant?
- In the methods section: the paragraph “intensity averaging over macroscopic interaction volume” is trivial and could go in the supplementary information, in order to prepare space for the derived analytical model, that, in my opinion, is more essential to present in the main part of the manuscript. Clarity and context: appropriateness of abstract, introduction and conclusions
- To really understand the underlying physics, the reader has to read the supplemental information (SI) of the paper, which features the analytical model. Many arguments of the main text are based on the theory derived in the SI. Since theory plays such a dominant role in this work, I would suggest to include a short summary of the analytical theory in the main text of the paper: The current theory summary of the main text “analytical model based on a Dyson series for the two-level system undergoing Rabi oscillations” does not reveal much and should be expanded. This could be done following equation (1) of the main text – here one could explain that on top of this solution, time-dependent perturbation theory is applied in order to drive the photoelectron spectra.
- The explicit mentioning of the strong-field approximations (references 19 and 20) reads somehow arbitrary in the conclusions, considering the fact that there are other previous works (for example reference 22) that show theoretical results beyond this approximation. I would not consider the theoretical method development itself as a strongpoint of the presented work. The theory is essential to support the interpretation of the data, but I do not consider it as a unique selling point of the paper. I would it deem interesting for the future reader to hear about further extensions/applications of the probing technique, or on future experimental opportunities that are enabled by the findings of this work.
- I would recommend the authors to sharpen their conclusions & outlook (rather than just giving a summary of the results) and to speculate about the generality of the observed quantum interference probe-technique.

Hamburg, 03-17-22 Nina Rohringer

Referee #3 (Remarks to the Author):

The manuscript by Nandi et al. reports observation of strong coupling of an XUV transition in Helium based on irradiation with seeded FEL pulses. The main point of the paper in my view is that with the advent of seeded FELs, it is possible to get coherent XUV pulses which are much shorter than the dissipation of atomic levels (here about 50 fs vs. 1 ns respectively; ignoring photoionization losses). Moreover, that such pulses can be sufficiently intense so that the Rabi period is similar to the pulse duration (here both are about 50 fs).

The authors probe these Rabi oscillations through the ejected photoelectrons in the pumping process. The lower level (a) can photoionize by two-photon absorption, while the upper can photoionize (b) due to single-photon absorption. As the authors mention, because a and b are coupled, the two-photon ionization can destructively interfere with the one-photon ionization, leading to Autler-Townes splitting. Moreover, the kinetic energy of the ejected photoelectrons is modified by the Rabi splitting of the upper- and lower-polaritonic states.

Generally speaking, I believe that the main point of this paper is important and the experimental results are new. Seeded FELs have been under development for quite some time, and this paper suggests that the capabilities are now present to probe light-matter interactions at XUV frequencies in much the same way that is done (with relative facility) at IR/Vis frequencies. While the paper could be suitable for Nature, I have some questions regarding the theoretical account of the experimental results that would should be answered, as well as some additional comments. They are presented in no particular order.

- The magnitude of the driving field is somewhat important for giving the reader the ability to cross-check the magnitude of the Rabi period. Page 5 indicates that the intensity used in the TDCIS simulations (of about 20 TW/cm²) is obtained from the Rabi splitting, which was backed out of the AT splitting. Is there more direct support for this intensity from direct measurements of the seeded FEL output? (Or the FEL simulations quoted on page 18). The authors should specifically state a measured or estimated intensity (e.g., given the pulse energy mentioned on page 18).

- I found the claims surrounding Fig. 3 to be surprising. In particular, that the non-resonant contribution to two-photon absorption is comparable to, or even dominant, over the resonant two-photon absorption. The authors provide an argument using the relative strength of the dipole matrix elements for intermediate p states, and the inferred magnitude of the electric field from the observed AT splitting. The authors should directly calculate the rate of resonant and non-resonant two-photon absorption (e.g., from perturbation theory) and affirm these estimates – as once again, this is not a priori expected. Moreover, that this result is observed in Helium: the authors should comment on whether or not there are other experiments affirming this strong non-resonant two-photon absorption.

- A discussion of the expected (or known) magnitudes of resonant and non-resonant two-photon absorption is also important because they indicate the damping rates of a and b, which are important to assess the feasibility of strong coupling. I believe for a discussion of these strong coupling effects to be complete, the authors should provide an estimate of such rates directly and

compare them to the coupling strength.

o Additionally, on page 9 (in the conclusion), the authors note that the giant Coulomb wave interpretation enables them to “explain how Rabi oscillations can prevail ... despite photoionization losses from the neutral atom”. This is a key point: it leads the reader to wonder if the uncoupled loss rates of the states are in fact rather high, and whether or not the strong coupling is completely dependent on this quantum interference between one- and two-photon pathways. The authors ought to comment on this, because one might imagine that such interferences are not generic to matter systems, and this may limit the set of systems for which X-ray strong coupling can persist, even with highly coherent and intense X-ray pulses (this is not intended to be a negative point; I simply believe it will enhance the paper for the reader to understand the generality or “special-ness” of the experimental results here).

- I have some quibbles with respect to terminology in the abstract and introduction. Referring to these effects as “quantum optical” is misleading: the effect is fully understood without referring to quantization of the XUV field. The authors should remove the statement that such effects are trademarks of “quantum optics” and simply note that they are fundamental to the physics of atom-field interactions (or similar).

Author Rebuttals to Initial Comments:

Overview of changes:

New figure added to the main manuscript:

* **Fig. 1(c):** In response to the comments by Referee 1 and 3 about damping effects, such as photoionization from the Rabi cycling atom, we have prepared a new panel that provides an overview of the different domains of Rabi oscillations in helium for the 1s-4p transition. The plot shows all relevant times scales (including spontaneous emission, photoionization processes and Rabi periods) vs the intensity of the FEL pulse. The figure clearly demonstrates that our experiment is carried out in a regime with negligible damping effects. It also shows that the photoionization rates by one photon (I) and two photons (II) are comparable in magnitude due to the high intensity of the FEL pulses, which is one of the main findings of our work. The new figure is added as sub-figure (c) in Fig. 1 and it further serves as a roadmap for future experimental investigations in our outlook. Fig. 1(a) is modified to introduce panel (c) properly.

New references added to the main manuscript:

* **Ref. X** (Ref. 36 in the manuscript): Holt, C. R., Raymer, M. G. & Reinhardt, W. P. Time dependences of two-, three-, and four-photon ionization of atomic hydrogen in the ground 1^2S and metastable 2^2S states *Phys. Rev. A* **27**, 2971 (1983).

* **Ref. Y** (Ref. 37 in the manuscript): Rzaewski, K., Zakrzewski, J., Lewenstein, M. & Haus, J. W. Strong-field autoionization by smooth laser pulses. *Phys. Rev. A* **31**, 2995 (1985).

* **Ref. Q** (Ref. 31 in the manuscript): Žitnik, M. et al., Lifetimes of n^1P states in helium. *J. Phys. B: At. Mol. Opt. Phys.* **36**, 4175 (2003).

* **Ref. Z** (Ref. 34 in the manuscript): Eberly, J. H. Area theorem rederived. *Opt. Exp.* **2**, 173 (1998).

Additional figures prepared for the rebuttal (not included in main text or, Methods/SI):

* **Fig. R1:** Rabi oscillation populations of $|a\rangle$ and $|b\rangle$ computed with TDCIS for helium atoms that are subjected to pulses with identical parameters estimated from the experiment. The result shows that Rabi oscillations take place with close to 100% modulation of populations. Furthermore, the photoionization loss is found to be $\sim 0.1\%$, which shows that the experiment can be interpreted as an *in-situ* experiment. This theoretical simulation, proposed by Referee 1, is a conclusive evidence for Rabi oscillations without any damping in our experiment.

* **Fig. R2:** Photoelectron spectrum computed with TDCIS for helium atoms that are subjected to pulses with increased intensity as suggested by the Referee 1 as a further test case. In accordance with our interpretation of the experiment, we observe that the Autler Townes (AT) doublet spacing increases with increasing intensity. The loss of contrast of the AT doublet is a trivial effect, as pointed out by Referee 2, and it explains why the “dip” between the two AT peaks in the experiment is not as deep as in this single atom simulation. Detailed discussions about the macroscopic effects are moved into the SI.

Referees' comments:

Referee #1 (Remarks to the Author):

In their manuscript, the authors describe an experiment on the dynamics of helium under the action of a seeded free electron laser in the XUV frequency regime. The focus is on Rabi oscillations, i.e., coherent periodic modulations of the occupation of two quantum states. The authors state that so far, such oscillations induced by short wavelength radiation have not yet been directly observed. Their main claim is that they induce such oscillations using XUV pulses, and observe them via the photoemission of electrons.

First of all, I am not convinced about the significance and novelty of the results. The introductory paragraph suggest the lack of a direct observation of Rabi oscillations at short wavelengths as the open challenge to be addressed. However, I do not think that the present experiment fills this gap. In my view it is not justified to speak of a direct observation because the entire analysis relies on heavy theory input. Without the comparison of complicated signatures to an extended theory model, it is impossible to pinpoint the existence of oscillations.

Our experiment consists of direct measurements of Rabi oscillations, not in the time domain, but in the energy domain. The observation of an Autler-Townes doublet structure is a direct signature of Rabi oscillations, as we show in our manuscript. The interpretation of our experimental results does not rely “on heavy theory input”, because the signal from the excited state, $|b\rangle$, is simply the Fourier transform of the excited state amplitude, $b(t)$. When the amplitude oscillates, according to the Rabi-theory the signal in the energy domain splits into a doublet structure. No “extended theory model” is required for drawing this solid conclusion out of our experimental observation.

To clarify that no heavy theory input is needed, after

“This is in agreement with the results from the analytical model presented in Fig. 1b.”

we now add:

“The contribution from the excited state, $|b\rangle$, can be understood as the Fourier transform of the time-dependent amplitude: $b(t)$. Similarly, the contribution from the ground state, $|a\rangle$, can be related to the Fourier transform of $a(t)$ via a non-resonant wave packet composed of complement states, $|c\rangle$, as illustrated in Fig.1a. Thus, the observation of a doublet from either state: $|a\rangle$ or $|b\rangle$ is a direct measurement of Rabi dynamics in the energy domain.”

The agreement between theory and experiment, e.g., in Fig 2, is only qualitative.

In the original version of the manuscript, in the Methods section we explained that a macroscopic sample inevitably leads to the loss of contrast for the photoelectron lines. In addition, we now explicitly mention this effect in the caption of Figure 2:

“The loss of contrast observed in the experimental spectra arises due to macroscopic averaging of the target gas sample (see SI for details).”

As pointed out by Referee 2, these are well-known effects and, thus, we decided to move the discussion about macroscopic averaging from Methods to the Supplementary Information.

The avoided crossing (Fig 4) is (seemingly) simpler to interpret, but it does not show any appreciable population transfer.

In Fig. R1, we show populations for the $|a\rangle$ and $|b\rangle$ states extracted from TDCIS simulations with the actual experimental parameters. Simulations were performed in velocity gauge, which is justified by the fact that the ponderomotive energy is small (around 5 meV), despite the large intensity of the FEL pulse. **The figure clearly shows the expected 1.5 Rabi cycle with close to 100% modulation of the populations.**

Fig. R1: (a) Ground state population: $|a(t)|^2$ and (b) excited state population: $|b(t)|^2$. The data is obtained by *ab initio* TDCIS calculations for a helium atom subjected to a time-dependent electric field having the same parameters as that of the FEL-pulse from the experiment (with a Gaussian pulse envelope). A clear ~ 1.5 Rabi oscillation is observed with close to 100% modulation depth in both the ground and excited state. The populations have been converged in all numerical parameters (in the figure we show convergence in the size of the radial box, by plotting simulations for different numbers of B-splines).

The amount of photoionization can be estimated using the data from Fig. R1 by subtracting the sum of the two final populations from the total population:

$$P_{\text{ion}} = 1 - (|a_f|^2 + |b_f|^2) = 0.1\%.$$

Since ionization-losses are negligible in the Rabi oscillations, **this shows that measurement of photoelectrons is an in-situ probe.**

However, the reported results do go beyond what has been reported before in that the claimed oscillations are induced by the XUV field. Previous experiments also involved XUV fields, but employed lower-frequency (IR) fields to induce the Rabi oscillations. This is indeed a difference, but it is not clear to me in how far this opens up new avenues - the manuscript does not really comment on this question.

We sincerely thank the referee for their appraisal of our work. We firmly believe that the transition from long wavelength to short wavelength opens a range of new areas of research: field-driven dynamics to states with atom specificity in complex targets, such as molecules, selective population inversion of core excitations and well-targeted post-photoionization ion dynamics. To make this clearer, we have now expanded our outlook in the conclusion with references to previous theoretical works that dealt with complex Rabi oscillations restricted to strong photoionization [Ref. X] and autoionization processes [Ref. Y].

To clarify how our findings open up new avenues, the following portion in our conclusions:

“Together with our experimental approach of using two-photon ionization as an in situ probe of the coherent population transfer, which does not rely on any additional laser probe field, the scheme proposed here becomes applicable to other quantum systems as well.”

is expanded to:

“Our experimental approach of using photoionization as an in situ probe of Rabi dynamics does not rely on any additional laser probe field, and hence, is easily applicable to other quantum systems. We think that such photoionization processes are of interest to different domains: the one-photon domain connected to the symmetric excited-state dynamics (I) and the two-photon domain connected to the asymmetric ground-state dynamics (II), as shown in Fig. 1 (c). At the boundary between these domains, quantum interference is manifested in the photoelectron signal. The experimental blueshift of the symmetric AT doublet, reported in our work, is a first observation of this type of effect. Due to substantially different angular distributions from domains I and II (see Extended Data Table 1), we predict that the boundary regime will further exhibit intricate angular dependencies of the photoelectrons that could be the subject of upcoming experiments. Studying more complicated Rabi dynamics, affected by rapid photoionization³⁶ or autoionization decay³⁷, are natural extensions of our work. Coherent population inversion with core-excited states and ultrafast core ionization with simultaneous ionic state excitation²¹ are examples of configurations achievable at short wavelengths. Given the ongoing developments of seeded FEL facilities around the world capable of providing light pulses down to few-angstrom wavelength, our findings can inspire future studies involving coherent control in multi-electronic targets, such as molecules, and nano-objects with site specificity, opening up new pathways for steering the outcomes of photo-induced processes across ultrafast timescales.”

I also fail to see how the results generate new understanding of the XUV-induced dynamics. The basic condition for Rabi oscillations is to have coherence- and population losses low enough such that the coherent dynamics is faster than the incoherent one. In this regard, there is no difference between the original Rabi model and the present paper. The challenge at higher photon energies only is that the typically required Rabi frequencies can not be realized without simultaneously inducing strong loss processes, such that a delicate balance is required. But this, of course, is also long known.

We agree with the referee that our work does not improve the understanding of the Rabi model, which is an excellent approximation of the dynamics of a two-level system in the weak-coupling regime. However, the **new physics, which in our opinion is quite remarkable, is that the one photon ionization from the excited state and the non-resonant two-photon ionization from the ground state have similar magnitudes due to the high intensity of the FEL pulse.** The

losses due to photoionization are negligible, so that Rabi oscillations can take place without damping, as clearly shown in Fig. R1.

I am even tempted to challenge whether "Rabi oscillations" have been observed at all. There are several reasons for this:

This issue is fully addressed with the clear time-domain Rabi oscillations shown in Fig. R1.

- The observation of oscillations requires that the coherence between ground and excited state is maintained throughout the dynamics. The authors argue that this is possible since the life time of the excited state is 1ns, much longer than the time scale of the experiment governed by the 50fs pulses. But this comparison is misleading, in my view. First of all, obviously the 1ns lifetime only is valid in the absence of external fields. In the presence of the fields, the photo ionization both, of the ground and excited state, like reduces the lifetime of the coherence also to the ~10fs scale or even below. In this situation, it is not clear if the oscillations persist, or if only a "damped" transfer from the ground to the excited state takes place. The authors do not comment on this issue. They also do not comment on the challenge that it may not be possible to go to higher Rabi frequencies (i.e., more oscillations) due to the inevitable related increasing ionization, but instead praise their main source of decoherence and population loss (ionization) as an in-situ probe.

In the original version of the manuscript, we explained that photoionization serves as an in situ probe of Rabi dynamics. The lifetimes for photoionization in our experiment are now estimated to be ~100 picoseconds. We think this is an important point raised by the referee and we therefore make the following changes to the manuscript:

The manuscript is updated with a new panel in Fig. 1 [Fig. 1(c)] that includes photoionization lifetimes for the atom undergoing Rabi cycling along with the associated Rabi period and our experiment marked with a diamond. Our experiment was conducted on the boundary between one-photon ionization (I) from $|b\rangle$ and non-resonant two-photon ionization (II) from $|a\rangle$. It can also be seen in Fig. 1(c) that our femtosecond FEL pulses are close to three orders of magnitude shorter than the photoionization lifetime of the atom (these estimates from the analytical model are consistent with *ab initio* TDCIS simulations: $1 - \exp(-1/1000) = 0.1\%$). Thus, decay processes due to photoionization can be safely neglected.

The new subpanel of Figure 1 is introduced in the main text by rephrasing:

“It is possible to drive the transition coherently, because the excited state lifetime (~ 1 ns) is much longer than the estimated full-width-at-half-maximum (FWHM) of the FEL pulse duration: 56 ± 13 fs.”

to read:

“As shown in Fig. 1c, several physical effects can impede the duration of Rabi cycling: spontaneous emission sets a fundamental limit at ~ 4 ns³¹, while one-photon ionization from the excited state (I) and two-photon non-resonant ionization from the ground state (II) restrict Rabi cycling at progressively higher intensities. The present experiment (diamond) can be driven coherently, as it takes place over an ultrafast time duration, with the estimated full-width-at-half-maximum (FWHM) of the FEL pulse duration being 56 ± 13 fs. This is three orders of magnitude shorter than the lifetime for photoionization of the Rabi cycling atom: $\tau_{a+b} \approx 100$ ps (see Supplementary Information for details).”

Fig. 1c: Domains of photoionization from $1s$ - $4p$ Rabi cycling helium atoms with the dominant photon process: I (red shaded-area) and II (blue shaded-area). Rabi cycling is limited by spontaneous emission (τ_{SE}), I-photon ionization from $|b\rangle$ (τ_b) and II-photon ionization from $|a\rangle$ (τ_a) at progressively higher intensities of the FEL field. τ_{a+b} is the lifetime of the Rabi cycling atom subject to photoionization. The boundary between the two domains is determined by $\tau_a = \tau_b$. The diamond marks the experiment: on the boundary between the I-II domains and close to a single Rabi cycle.

In addition, we have added the following text to the description of the model *in the SI*:

“From the ionization amplitudes it is possible to derive an effective ionization rate that takes into account the contributions from the one- and two-photon processes. For simplicity, we assume resonant photon energy, and that the pulse is long enough to form a well-separated Autler-Townes doublet in the spectrum. Following the procedure that gives rise to Fermi’s

golden rule (see e.g. *Modern Quantum Mechanics* by J. J. Sakurai) leads to an ionization rate for each partial wave l that is the sum of the rates of ionization from $|a\rangle$ and $|b\rangle$,

$$\Gamma_{a+b}^l = \Gamma_b^l + \Gamma_a^l = \pi \left(\left| \frac{z_b^l E_0}{2} \right|^2 + \left| \frac{z_{\rho \neq b}^l E_0^2}{4} \right|^2 \right),$$

where the individual rate from each state is reduced by a factor 1/2 compared to a straightforward application of the golden rule. This factor 1/2 is understandable, since at resonance each of the two states is populated only half of the time, and is consistent with what has previously been reported for the one-photon process [Ref. X]. From the effective rate, we get an estimate of the lifetime as $\tau_{a+b} = 1/\Gamma_{a+b}$, where $\Gamma_{a+b} = \sum_l \Gamma_{a+b}^l$ is the sum of the rates for all partial waves for photoionization from both $|a\rangle$ and $|b\rangle$. The lifetime is plotted together with the individual contributions from $|a\rangle$ and $|b\rangle$, denoted τ_a and τ_b , in Fig. 1(c) of the main manuscript. From TDCIS simulations, we find that the ionization yield in the experiment is within 0.1%. This is consistent with the lifetime estimate from the model (~100 ps) which gives an ionization yield close to 0.05% depending on the exact pulse length.”

Second, at least in the traditional Rabi model, the lifetime of the upper state is not really an issue by itself. Rather, the ratio of the Rabi frequency to the loss rates should be the key quantity which determines whether oscillations appear or not? But the authors do not comment on the Rabi frequency in their qualitative discussion on page 4.

As shown in Fig. R1 and Fig. 1(c), photoionization losses are negligible (0.1%) and Rabi oscillations are close to 100% in population modulation. As pointed out previously in our reply, the Rabi frequency (~52 fs) is faster by almost three orders of magnitude compared to the photoionization losses (~100 ps), which we have now explicitly mentioned in the manuscript.

- Furthermore, the archetype two-level system of the Rabi model is strongly modified in the present experiment by other states coupled by the applied field - see Fig. 3(a). The interpretation of the photoelectron spectra in Fig 2 is only possible if this extended level scheme is considered. The authors stress the significance of other excited states (with the "giant wave" which is not really explained in the manuscript in a way that would make it accessible to a more general reader, let alone its effect on the dynamics in the a/b subspace). Again, it is not clear if there are really oscillations between $|a\rangle$ and $|b\rangle$.

We now clarify the “giant wave” concept by rewriting:

“due to constructive addition of non-resonant intermediate p -states (see grey bound and continuum states in Fig. 3a). This leads to a giant localized wave: $|\rho_{\neq b}\rangle$, in comparison with the normalized wavefunction for $|b\rangle$, as shown in Fig. 3b.”

as:

“due to superposition of intermediate (complement) states, which are illustrated as gray bound and continuum states in Fig. 3a. This leads to a giant localized wave: $|\rho_{\neq b}\rangle$, in comparison with the normalized wavefunction for $|b\rangle$, as shown in Fig. 3b. The largest contributions to the giant wave come from the dipole-allowed complement states that are close to the one-photon excitation energy.”

In the main text, we explained the physical interpretation of the “giant wave”. We show the complement states contributing in Fig. 1(a) and 3(a), as well as the physical form of the giant wave in Fig. 3 (b). With this, we believe that the general reader will now be satisfied with the explanation provided in the main text itself.

- Only after having discussed Rabi oscillations in detail, the authors state on page 5 that their experiment is close to a single Rabi cycle. I am not sure if the majority of readers would associate this to "Rabi oscillations" promised in the introduction. Note that the short pulse duration does not restrict the number of Rabi cycles a priori because in principle one could resort to higher Rabi frequencies than what has been achieved in the manuscript.

We assume that the referee’s objection is to the *plural* form of Rabi oscillations. Therefore, we change:

“oscillations”

to

“dynamics”

to avoid possible confusion in the introduction of the manuscript.

* The photoelectron kinetic energies in Fig. 4 even suggest that there is no appreciable excitation of the upper state $|b\rangle$ at all. There is no second resonance associated to the dressed state asymptotically belonging to $|b,0\rangle$. The authors write "This is because the interaction there is weak and the atom remains mostly in its ground state". **This again raises the question whether oscillations really have been observed.** Further, is it really justified to speak about dressed states in this situation? If one diagonalizes the 2-level Hamiltonian, then both eigenstates should contain significant population even in the limit of low excitation?

Concerning this question about whether Rabi dynamics take place or not, we refer to Fig. R1 and Fig. 1(c) and stress that photoionization losses are negligible.

In our manuscript we do explain that $b(t)$ has two symmetric energy components, while $a(t)$ has two asymmetric energy components. Asymptotically, far from the resonance, the contributions of the excited state decrease as: $\sim \pm\Omega/W \rightarrow \Omega/\Delta\omega \rightarrow 0$. In contrast, the contribution of the ground state becomes: $\sim (1 \pm \Delta\omega/W) \rightarrow 2$ or 0. This means that at *very* large detuning only a single

peak should be observed. This is consistent with our experimental, numerical and model observations.

However, close to the resonance the photoelectron signal is sensitive to quantum interference between the one-photon domain (I) and the two-photon domain (II) and this is the reason for the sharp exclusion of the $|b,0\rangle$ line. The explanation can be found by studying Eq. S8 and S11, where different terms have different signs that act to enhance *one of the peaks* beyond the asymmetric nature from the ground state amplitude. We anticipate that future work will be aimed at understanding the transition from (I) to (II), where rich interference effects and angular distributions can be studied to understand the underlying dynamics.

In order to clarify the dependence of one- and two-photon processes on detuning, we change the sentence:

“This is because the interaction there is weak and the atom remains mostly in its ground state, $|a\rangle$, such that two photons are required for photoionization.”

to read:

“This is because the interaction is weak far from the resonance with the atom remaining mostly in its ground state $|a\rangle$, such that two photons are required for photoionization. The region closer to the resonance is influenced by the quantum interference between the one- (I) and two-photon (II) processes that leads to further suppression of $|b,0\rangle$ and enhancement of $|a,1\rangle$, with both coupled energies appearing briefly to form an avoided crossing in kinetic energy.”

- The manuscript does not relate to traditional key signatures of Rabi oscillations, or experimental approaches to verify them. For example, a typical approach would be to vary the field intensity, while keeping the field on resonance. This way, the change of the oscillation period as function of time could be observed. Of course, time-resolved measurements are challenging with fs pulses.

Future experiments will likely be aimed at time-domain measurements of Rabi oscillations, but such experiments would be even more challenging than our experiment. **We have chosen a more pragmatic approach to detect Rabi oscillations, which we believe will inspire novel experiments on Rabi-oscillating quantum systems at short wavelengths.** We note that a time-resolved experiment using narrowband coherent XUV-pump + XUV-probe to study photoionization processes has not been achieved yet. While laboratory-based tabletop HHG-sources are progressing towards that direction, these pulse trains are still broadband, leading to excitation of many levels at the same time (see, e.g., T. Okino et al., *Science Advances* **1**, 1500356 (2015)). In that sense, our technique circumvents the current experimental limitations in an efficient way.

But a variation of the pulse intensity would be feasible, and from the interpretation given by the authors in Fig 1, it should allow one to observe qualitatively different photoelectron spectra as function of the pulse area.

We have performed similar simulations and an example is presented here. The key result is that the Autler-Townes doublet widens when the intensity of the FEL pulse is increased: $\Omega \sim \sqrt{I}$. Our conclusions are fully consistent with these observations:

Fig. R2: Photoelectron energy distributions computed by *ab initio* TDCIS simulations for helium atoms. Each curve corresponds to a new simulation with the intensity specified on the y-axis, and the color and shape of the lines correspond to photoelectron density. Increasing the intensity of the FEL pulse leads to a widening of the Autler-Townes doublet, which is equivalent to an increased Rabi frequency. The peaks are computed with FEL frequency tuned to the $1s-4p$ transition (and **not** to the symmetric doublet condition). This explains why the two peaks are not of equal strength. Additional smaller sub-peaks appear when multiple Rabi oscillations take place (see for instance Zhang, S. B. and Rohringer, N. *Phys. Rev. A* **89**, 013407 (2014)).

Instead, the experiment varies the detuning of the field. This also does change the *generalized* Rabi frequency $\sqrt{\Delta^2 + \Omega^2}$ and thus the oscillation frequency. But at the same time, it also modifies the oscillation amplitude, even in the ideal case, and potentially also relative phases between the different ionization channels are modified on the $1/\Delta$ timescale, which are not accounted for in the interpretation in Fig 1. As a result, the detection is modified as well. I do not think that the present experimental results do allow one to disentangle the effects of the actual desired dynamics (Rabi oscillations) from those of the modified detection.

Indeed, the fact that the relative phases between the one- (I) and two-photon (II) ionization processes vary with detuning, has led us to discover **a previously unknown phenomenon: a blueshift of the Autler-Townes doublet of photoelectrons from Rabi oscillating atoms**. This is one of the major experimental results of our work and we have explained that this phenomenon is the conclusive evidence for interference effects between the one- (I) and two-photon (II) processes in our manuscript. In Fig. 1(b) and 2(c,d) we introduce the reader to the two contributions [(I) and (II)] separately, without any interference effects. Please note that Fig. 2(d) shows smaller peaks of similar height on both sides of the detuning, while the *ab initio* simulation shows smaller peaks with slightly different heights (see Fig. 2(b)). The reason for this is the quantum interference between pathways I and II. In this regard, the use of *ab initio* simulations, which do contain the relevant phases, has been used appropriately to study and validate the interference between the two processes. Finally, looking at the experiment in Fig. 2(a), the smaller side-peaks (or “shoulders” due to macroscopic averaging) show a similar trend with the red side-peak being stronger than the blue side-peak. This also clearly demonstrates that the experiment is sensitive to the quantum interference between I and II.

For example, what is the meaning of an atom performing $3/2$ Rabi oscillations if it continuously undergoes ionization on similar time scales?

Or, would it be possible that the line shape changes in Fig 2 are not due to Rabi oscillations /AT splitting, but due to varying phases induced by the detuning of different ionization channels, like in Fano resonances? (One should also note that the energy of the symmetric configuration differs between experiment and theory.)

We refer to Fig. R1 and Fig. 1(c) and emphasize that photoionization losses are negligible in our experiment. As mentioned above, the *ab initio* simulations indeed took into account the appropriate phases of the different ionization channels to validate the observed quantum interference phenomenon. As already explained in the manuscript, the energy of the symmetric configuration being different between theory and experiment is due to the correlation effects not included in TDCIS that increases the binding energies.

- The experiment is in an entirely different regime than traditional Rabi flopping experiments, due to the highly transient nature. This is not necessarily a disadvantage, but one may wonder in how far a direct relation to the original Rabi model is justified. For example, it is reported that the Rabi dynamics in the present setup is sensitive to the exact shape of the driving pulse. On the contrary, in the resonantly driven original Rabi model, the dynamics only depends on the pulse area (area theorem), and not on the pulse shape. Further, the Rabi frequency - and with it the AT splitting - is highly transient in the present experiment. Why would one still expect a splitting as shown in Fig 2 or in Fig 4, even though the ionizing field is always present in the experiment (unlike in previous work which relied on a time-delayed ionizing pulse)? Also, the AT splitting should be time-dependent as well, ranging from 0 to the maximum value.

The area theorem is useful to explain the number of Rabi cycles from a given pulse. In fact, the scaling of $3/2$ that we reported between flat-top and Gaussian pulses comes from the area theorem. We add to the sentence:

“For instance, a Gaussian pulse can induce more Rabi oscillations than a flat-top pulse with the same FWHM by a factor of $\sqrt{\pi/2\ln 2} = 1.5\dots$ ”

“... as follows from the area theorem³⁴.”

in the main text for clarity.

The present setup is sensitive to the exact pulse form because it is tuned to the boundary between domain I and II. We think that this dynamical sensitivity to the exact pulse that comes from dynamics outside the two-level system is an important finding of our work, as emphasized in the current manuscript.

We agree that “the AT splitting should be time-dependent as well, ranging from 0 to the maximum value”. It is now shown *schematically* in the updated version of Fig. 1(a).

On a more technical side, the shape of the experimental and the theoretical curves in Fig 2 are somewhat different. In particular, there are pronounced "dips" in the theoretical curves which are not present in the experimental data. What is the reason for this? If the dips are due to interferences, then the missing dips could be interpreted as arising from (partially) incoherent dynamics.

The “dips” in the experiment, shown in Fig. 2(a), are weaker than the “dips” in the theory, shown in Fig. 2(b-d), because of macroscopic averaging of the atomic gas sample in the experiment. We have commented on this effect earlier in our reply.

Since the analysis relies on heavy theory anyway, I wonder whether it would be possible to calculate the actual dynamics projected into the $|a\rangle$, $|b\rangle$ subspace for the experimentally relevant parameters. The normalization should be such that 100% refers to the initial population in the ground state. This way, the population losses out of the two-level system, and the presence of oscillations (or damped dynamics) could be judged. This may not be completely reliable since the theory predictions differ somewhat from the experimental data, but if no clear oscillations would be visible in the theory, then one should safely be able to reject the notion of population oscillations in the experiment.

We wish to refer to Fig. R1 and Fig. 1(c) and highlight that photoionization losses are negligible. We disagree with the referee’s claim that “the analysis relies on heavy theory anyway”, since the measurements were carried out in the energy domain with the Autler-Townes doublet being a direct indication of Rabi oscillations.

Regarding the experiment, is there sufficient data at different pulse intensities but fixed detuning? Would this allow for an analysis of the dependence of the Rabi oscillations on the pulse area/peak intensity? This could form a more direct signature of an oscillation.

We did not measure the “data at different pulse intensities but fixed detuning”. The experiment was performed without any metallic filter in FEL-beam path, meaning it was the highest possible intensity experimentally achievable at this photon energy, without introducing any distortion in the spectral shape of the pulse. As pointed out in Fig. 1(c), our measurement resides in a region where quantum interference between pathways I and II plays an important role in forming the blueshift of the Autler-Townes doublet. Strong reduction in intensity via metallic filters would hinder the observation of such an effect, which, in turn, is one of the main findings of our work.

Overall, I cannot recommend publication of the manuscript in Nature for the above reasons.

We are hopeful that with this detailed response that includes:

- * a simple explanation of how **our results can be understood as direct measurements of Rabi oscillations** in the energy domain,

- *several **additional simulations proposed by the referee** (Fig. R1 and Fig. R2) to show explicitly that Rabi oscillations are taking place in the atom with close to 100% modulation of populations,

- * **an updated manuscript** with clarifications about our model, interpretation and outlook (please note the changes in main text marked in blue in our reply to the other referees as well),

- * explicit **calculations for photoionization lifetimes** from atoms undergoing Rabi oscillations,

- * **a new panel: Fig. 1(c) that serves as a road-map and inspiration for future experiments,**

the referee will agree that the present work opens up a new research avenue for coherent light-matter interactions and, on the way, fills an important gap in our understanding of Rabi dynamics at XUV wavelengths, which warrants publication in *Nature*.

Referee #2 (Remarks to the Author):

The authors present photo-electron spectra of Helium driven with high-intensity XUV radiation close to the 1s to 4p resonance. The intensity of the XUV pulses is high enough to induce one to two Rabi Oscillations between the ground and excited state during the duration of the pulse. The post-processed spectra show a splitting of the photo emission line, reminiscent of Autler-Townes doublets. Moreover, the evolution of the spectrum as a function of detuning shows a feature hinting towards an avoided crossing. The interpretation of the experimental relies on a substantial theory effort: Time-dependent Configuration Interaction Singles (TDCIS) calculations qualitatively reproduce the features of the experiment. Moreover, an analytical model based on time-dependent second-order perturbation theory on top of the two-level Rabi dynamics is presented and serves as the basis of the interpretation of the experimental photoelectron data. The measured line shapes are explained as a dynamical interference effect between a 1-photon ionization pathway from the excited state and a 2-photon ionization from the ground state. Interestingly, the authors identified an effective intermediate state in the 2-photon transition featuring a giant effective dipole moment that leverages the generally weak 2-photon process. The arguments and supporting theory fully corroborate the interpretation of the line shapes in terms of Rabi Oscillations. All the appropriate credit has been given and previous work was cited accordingly. This work demonstrates a wonderful interplay between theory and experiment.

We sincerely thank the referee for her positive assessment of our work.

Originality and significance: The authors could, for the first time, demonstrate Rabi Oscillations in EUV bound-to-bound transitions, by studying their manifestation in photoelectron spectra. These kind of experiments are extremely difficult to achieve at short-wavelength free-electron laser sources. By using the seeded FERMI FEL, previous limitations to show the effect of Rabi oscillations at SASE FELs of intrinsically limited temporal coherence could be mitigated. The discovered dynamical interference pathway of 1- and 2-photon transitions of the driven atomic systems could in the future eventually, if set-up in pump-probe schemes, develop into powerful phase-sensitive tool to study electron-wave packet dynamics. It remains questionable, if the effect of the “giant intermediate wave”, that is necessary to observe strong coherence effects, is a general feature, or particular to Helium.

We have verified that the giant wave phenomenon is not unique to the 1s-4p transition in helium because we have studied other couplings in the $1s-np$ Rydberg series (not shown). It is reasonable to assume that similar enhancement effects will occur in general valence to Rydberg coupling in other atoms as well. As shown in Fig. 1(c), the photoionization lifetimes of one-photon ionization from the excited state (I) and two-photon non-resonant ionization from the ground state (II) will cross in the space of intensity and time. However, other decay mechanisms, such as spontaneous emission, super fluorescence, autoionization or decoherence due to coupling

to other degrees of freedom in a complex molecule, should also be considered in future extensions of this scheme. See the updated outlook and Fig. 1(c) in our manuscript.

Data & methodology: The authors did a careful data analysis, transparently presented in the manuscript. Moreover, two distinct theory results have been presented – a time-dependent configuration interaction calculation and an analytical model, the basic building block on which the interpretation of the experimental findings rests.

Appropriate use of statistics and treatment of uncertainties:

- In order to get an idea about the standard deviation of the experimental data, it would recommend the authors to include according bars in Extended Data Figure 4. Does the deconvolution algorithm allow to also assign error bars to the processed data?

We have now added errors in the raw data shown in Extended Data Fig. 4, assuming Poisson counting statistics (for N -counts the corresponding Poisson fluctuations: \sqrt{N}). The deconvolution algorithm does not allow propagation of error. Future work will incorporate it for photoelectron spectra that are more complex. We have added the following sentence in the caption of Extended Fig. 4:

“The shaded region in each sub-panel represents the corresponding Poisson fluctuations of the photoelectron signal.”

We wish to mention that the reported value of the Autler-Townes splitting: 80 ± 2 meV, was obtained after fitting the processed data with two Voigt profiles corresponding to the symmetric splitting. The error bars corresponding to the value of AT-splitting is solely due to this fitting procedure. We have now added the following sentence following the reported value in the main text:

“The reported uncertainty is obtained from a fit of the symmetric AT doublet with two Voigt profiles having same width.”

- Figure 3 c and d: A false-color scale should be added. Is the color intensity shown in the figures according to a linear or logarithmic scale? A logarithmic scale could maybe enhance the visibility of the additional Autler-Townes peaks that are visible in d).

We thank the referee for this instructive comment. We have now added the false-color scale to Fig. 3(c) and (d). The scale is linear, because using a logarithmic scale would enhance the wings observed in Fig. 3(d) that originates from the use of a flat-top pulse (‘sinc’ function in the frequency domain) in the analytical model.

- Figure 4 b-d: I suggest the same improvements as for 3 c and d

We thank again the referee for this instructive comment. We have included the false-color scale in all three panels, from Fig. 4(b) to (d). As before, we did not use logarithmic scale, as it will enhance the wings resulting from the use of a flat-top pulse in Fig. 4(d).

We have also added the false-color scale to Extended Data Fig. 3.

Suggested minor improvements:

- At several points in the main text (page 5 line 73 and page 7 line 118, page 8 line 140,) the authors refer to the “required” blue shift of the applied frequency in order to produce symmetric photoelectron lineshapes/position of avoided crossing. Why mention this fact twice? What can be learned from this shift? Why is it important? Can one experimentally determine a relative value of the effective transition matrix elements/cross sections of one- versus two-photon ionization?

The blueshift is one of the main findings in our experimental work and we have shown that it indicates quantum interference between one-photon (I) and two-photon (II) pathways. We have changed the manuscript to make this clearer.

After:

“A slight blue detuning of the XUV light by ~ 11 meV, relative to the atomic transition, is required to record a symmetric AT doublet (black squares in Fig. 2a).”

we add:

“This blueshift is one of the major experimental results and identifying its physical origin is among the main objective of the theoretical efforts presented in this work.”

We further rephrase:

“The required blueshift for the symmetric AT doublet is the signature of an interplay between the one-photon and two-photon processes that depends on the exact pulse form.”

to:

“Thus, we have found that the blueshift of the symmetric AT doublet is due to quantum interference between the one-photon (I) and two-photon (II) processes.”

After:

“The enhanced shift of the AT doublet to blue detuning in the analytical model is an effect of the pulse-envelope that can be reproduced with TDCIS using smoothed flat-top pulses.”

we add:

“This indicates that the amount of blueshift of the AT doublet can be coherently controlled by the exact profile of the ultrashort FEL pulse.”

- The title “Studying ultrafast Rabi dynamics with short-wavelength seeded free-electron laser” might be a bit misleading, since no time-dependent measurements have been made. I would suggest to replace “dynamics” with “oscillations” in the title.

We believe that the title that we have chosen is slightly more general than the one suggested by the referee, because **we not only detect the ultrafast oscillations of the two-level system, but also probe the non-linear photoionization dynamics of atoms undergoing Rabi cycling at short wavelengths**. In future works, at higher intensities, such photoionization dynamics in higher order of perturbation theory will also affect the dynamics in the two-level system by complex damping effects. This point is now emphasized in the manuscript in the updated outlook with Ref. [X]. Hence, we wish to keep the title unchanged.

- In the abstract (introductory paragraph) the authors refer to “The measured photoemission signal revealed...” Since it could be ion- or electron signal, I would suggest to be more specific and write “The measured photo-electron spectra revealed...” or similar referring to the photoelectron.

We agree.

“The measured photoemission signal”

is changed to:

“The measured photoelectron signal”

- In the introductory paragraph, instead of stating “using theoretical analyses that go beyond the strong-field approximation...” I would rather refer to the actual theory that has been developed, i.e. time-dependent perturbation theory on top of the Rabi dynamics, or similar.

We agree and make this change as suggested.

“Using theoretical analyses that go beyond the strong-field approximation, we found that the ultrafast build-up of the doublet structure follows from a quantum interference effect between resonant and non-resonant photoionization pathways.”

is changed to:

“Using an analytical model derived from perturbation theory on top of the Rabi model, we found that the ultrafast build-up of the doublet structure carries the signature of a quantum-interference effect between resonant and non-resonant photoionization pathways.”

- Line 77, page 5: I suppose that “quantitatively” should actually read “qualitatively”.

We agree and change it accordingly.

- Line 114, page 7: “to simulate angle-integrated measurements” The mentioning of an angular integral here is somehow ad hoc and there should not be much angular dependence, starting out with a spherically symmetric He groundstate. Why referring to the angular integral here? Could it be that the average over the spatial interaction volume is meant?

We predict that the angular distributions will strongly depend on the atomic transition and FEL parameters due to the different angular distributions of the one (I) and two (II) photon processes (see Extended Data Table 1). Since this is an important point for future works, we have now added a sentence about this in the updated outlook in the manuscript.

- In the methods section: the paragraph “intensity averaging over macroscopic interaction volume” is trivial and could go in the supplementary information, in order to prepare space for the derived analytical model, that, in my opinion, is more essential to present in the main part of the manuscript.

We agree with the referee that the discussion about macroscopic averaging can be moved from the Methods to the end of the SI.

The fact that spectral features are broadened due to macroscopic averaging are now mentioned directly in the caption of Fig. 2 with reference to the SI.

Clarity and context: appropriateness of abstract, introduction and conclusions

- To really understand the underlying physics, the reader has to read the supplemental information (SI) of the paper, which features the analytical model. Many arguments of the main text are based on the theory derived in the SI. Since theory plays such a dominant role in this work, I would suggest to include a short summary of the analytical theory in the main text of the paper: The current theory summary of the main text “analytical model based on a Dyson series for the two-level system undergoing Rabi oscillations” does not reveal much and should be expanded. This could be done following equation (1) of the main text – here one could explain that on top of this solution, time-dependent perturbation theory is applied in order to drive the photoelectron spectra.

We thank the referee for this instructive comment.

“In order to interpret this non-linear dynamics, we have developed an analytical model based on a Dyson series for the two-level system undergoing Rabi oscillations (see Supplementary Information for details).”

is changed to:

“In order to interpret this nonlinear dynamics, we have derived an analytical model by partitioning the Hilbert space into the two-level system and its complement. We expand the time evolution in the form of a Dyson series, where in the zeroth order the two-level system undergoes Rabi oscillations. The photoionization dynamics from the excited (ground) state is treated by first (second) order time-dependent perturbation theory describing the FEL interaction with the complement of the two-level system (see Supplementary Information for details).”

- The explicit mentioning of the strong-field approximations (references 19 and 20) reads somehow arbitrary in the conclusions, considering the fact that there are other previous works (for example reference 22) that show theoretical results beyond this approximation.

Ref. 20 (or, Ref. 19 in the updated manuscript) reported that the strong-field approximation was insufficient to model the AT doublet quantitatively (SFA predicts that the contributions from $|b\rangle$ (I) are too strong), while Ref. 19 (or, Ref. 18 in the updated manuscript) pointed out that one photon-ionization was insufficient. In Ref. 20, conclusions were drawn by comparing SFA with *ab initio* simulations for hydrogen by propagation of the TDSE. Our model solves this issue for the present experimental parameters.

To avoid any room for ambiguous citation here we have removed Ref. 19 from this sentence. Furthermore, we explicitly mention Ref. 22 (or, Ref. 21 in the updated manuscript) in the updated outlook as a future experiment at seeded FELs.

I would not consider the theoretical method development itself as a strong point of the presented work. The theory is essential to support the interpretation of the data, but I do not consider it as a unique selling point of the paper. I would it deem interesting for the future reader to hear about further extensions/applications of the probing technique, or on future experimental opportunities that are enabled by the findings of this work.

- I would recommend the authors to sharpen their conclusions & outlook (rather than just giving a summary of the results) and to speculate about the generality of the observed quantum interference probe-technique.

We have now rephrased our conclusions putting less emphasis on the theoretical method development and added details about future extensions and applications (see the updated outlook in the manuscript). We thank the referee for encouraging us to use the outlook to give our view on future developments. In particular, we believe that site specificity in complex molecules for

coherent control and state preparation for super fluorescence at XUV wavelengths are promising future scientific directions.

Hamburg, 03-17-22 Nina Rohringer

Referee #3 (Remarks to the Author):

The manuscript by Nandi et al. reports observation of strong coupling of an XUV transition in Helium based on irradiation with seeded FEL pulses. The main point of the paper in my view is that with the advent of seeded FELs, it is possible to get coherent XUV pulses which are much shorter than the dissipation of atomic levels (here about 50 fs vs. 1 ns respectively; ignoring photoionization losses). Moreover, that such pulses can be sufficiently intense so that the Rabi period is similar to the pulse duration (here both are about 50 fs).

The authors probe these Rabi oscillations through the ejected photoelectrons in the pumping process. The lower level (a) can photoionize by two-photon absorption, while the upper can photoionize (b) due to single-photon absorption. As the authors mention, because a and b are coupled, the two-photon ionization can destructively interfere with the one-photon ionization, leading to Autler-Townes splitting. Moreover, the kinetic energy of the ejected photoelectrons is modified by the Rabi splitting of the upper- and lower-polaritonic states.

Generally speaking, I believe that the main point of this paper is important and the experimental results are new. Seeded FELs have been under development for quite some time, and this paper suggests that the capabilities are now present to probe light-matter interactions at XUV frequencies in much the same way that is done (with relative facility) at IR/Vis frequencies. While the paper could be suitable for Nature, I have some questions regarding the theoretical account of the experimental results that would should be answered, as well as some additional comments. They are presented in no particular order.

We sincerely thank the referee for their positive evaluation of our work.

- The magnitude of the driving field is somewhat important for giving the reader the ability to cross-check the magnitude of the Rabi period. Page 5 indicates that the intensity used in the TDCIS simulations (of about 20 TW/cm²) is obtained from the Rabi splitting, which was backed out of the AT splitting. Is there more direct support for this intensity from direct measurements of the seeded FEL output? (Or the FEL simulations quoted on page 18). The authors should specifically state a measured or estimated intensity (e.g., given the pulse energy mentioned on page 18).

Given the value of measured pulse energy at the output of the FEL undulator (87 μ J), the beamline transmission of about 30%, the estimated value of pulse duration (56 fs) and beam waist w_0 at the best focus (10.2 μ m), the estimated peak intensity is $\sim 1.4 \times 10^{14}$ W/cm². While it

proves that the FEL pulses were indeed intense enough to drive the Rabi dynamics, due to the shot-to-shot fluctuation of the FEL pulse duration observed in the experiment, this value could not reliably be used as the intensity input for the theoretical calculations. In addition, the Rabi dynamics depends on the area of the pulse and not on the peak intensity. Hence, the expectation value of max **E**-field from the average intensity would be the root-mean-square of the electric field rather than the mean of the electric field magnitude, which can over-estimate the electric field amplitude. We have now added the following sentences in the SI:

*“The peak intensity was estimated to be $\sim 1.4 \times 10^{14}$ W/cm². This clearly shows that the FEL pulses were intense enough to drive the coherent Rabi dynamics. However, this peak intensity alone does not directly correlate to the ultrafast Rabi dynamics, since the Rabi frequency is proportional to the **E**-field strength, and Rabi cycling requires sufficient area of the pulse for the AT doublet to emerge. Furthermore, given the shot-to-shot fluctuations of the FEL pulse parameters, we have used the observed AT splitting to extract the average interaction strength in a reliable manner without any need for simulated values of the pulse duration or experimental mean pulse energy.”*

- I found the claims surrounding Fig. 3 to be surprising. In particular, that the non-resonant contribution to two-photon absorption is comparable to, or even dominant, over the resonant two-photon absorption. The authors provide an argument using the relative strength of the dipole matrix elements for intermediate p states, and the inferred magnitude of the electric field from the observed AT splitting. The authors should directly calculate the rate of resonant and non-resonant two-photon absorption (e.g., from perturbation theory) and affirm these estimates – as once again, this is not a priori expected.

We thank the referee for this rigorous comment. We already performed simulations similar to the referee’s suggestions and provided the exact ratios for the model right after Equation S13 in the SI of our manuscript:

“At an intensity of 2×10^{13} W/cm², which corresponds to an electric field amplitude: $E_0 = \omega A_0 = 0.023880$ a.u., we estimate $R^s = 0.13546$ and $R^d = 1.1958$. While the one-photon transition is clearly dominant to the s-wave, the two contributions to the d-wave are more comparable in magnitude.”

Furthermore, the new panel, Fig. 1(c), shows the crossing between photoionization lifetimes as a function of FEL intensity. These lifetimes are related to the strength of the dipoles as: $\tau \sim 1/z^2$ and the expected crossing of the lifetimes can indeed be seen very close to the FEL intensity used in our experiment.

Moreover, that this result is observed in Helium: the authors should comment on whether or not there are other experiments affirming this strong non-resonant two-photon absorption.

- A discussion of the expected (or known) magnitudes of resonant and non-resonant two-photon absorption is also important because they indicate the damping rates of a and b , which are important to assess the feasibility of strong coupling. I believe for a discussion of these strong coupling effects to be complete, the authors should provide an estimate of such rates directly and compare them to the coupling strength.

We are unaware of any such experiments in the weak-coupling regime using non-resonant two-photon ionization. However, we have now cited a theoretical work, where related photoionization parameters for hydrogen were extracted from Floquet theory (Ref. [X]). It also presents a theory for complex Rabi oscillations, which we believe could be an interesting future direction for the research field.

We are not aware of any such magnitudes in the literature for helium atoms. For this reason, **we have now computed photoionization lifetimes of Rabi cycling atoms, with our model at the level of CIS theory, and added them to the new version of the manuscript in the form of a new panel, Fig. 1(c).** The lifetimes are much longer than the ultra-short FEL pulse duration ($\sim 1000:1$), which is also consistent with the low amount of photoionization (0.1%) that we observe in TDCIS simulations.

The referee mentions about the “strong coupling regime” and we would like to stress that our experiments are performed at the “weak coupling regime”, as we have written in our original manuscript. Still we observe almost 100% population transfer between $|a\rangle$ and $|b\rangle$ on an ultrafast time scale (see Fig. R1 in page 3 – 4 of the present document). In the “strong coupling regime” Rabi oscillations are no longer a good model and the Rabi frequency is close to the energy of the field free transition: $\Omega \approx \omega$. We think this is quite challenging to achieve experimentally using FEL pulses at XUV wavelengths, since it would imply a Rabi frequency of tens of eV. The detailed role of other states in the atom would enter before the strong-coupling regime is reached, which is beyond the scope of the present work.

o Additionally, on page 9 (in the conclusion), the authors note that the giant Coulomb wave interpretation enables them to “explain how Rabi oscillations can prevail ... despite photoionization losses from the neutral atom”. This is a key point: it leads the reader to wonder if the uncoupled loss rates of the states are in fact rather high, and whether or not the strong coupling is completely dependent on this quantum interference between one- and two-photon pathways.

We thank the referee for pointing out that this sentence could make the reader think that the photoionization rate was high, while it is very low (0.1%) and that photoionization constitutes an in situ measurement of the Rabi dynamics. See Fig. R1 in page 3 – 4 of in the present document for the assertion that photoionization losses are negligible. The new panel, Fig. 1(c), makes this point clear in the updated manuscript. We also rephrase:

“With this model, we now understand how Rabi oscillations can prevail at short wavelengths despite photoionization losses from the neutral atom.”

to :

“With this model, we now understand how ultra-fast Rabi dynamics at short wavelengths are imprinted on photoelectrons from weakly ionized atoms.”

The authors ought to comment on this, because one might imagine that such interferences are not generic to matter systems, and this may limit the set of systems for which X-ray strong coupling can persist, even with highly coherent and intense X-ray pulses (this is not intended to be a negative point; I simply believe it will enhance the paper for the reader to understand the generality or “special-ness” of the experimental results here).

We thank the referee for this instructive suggestion. The Autler-Townes doublet arises from both $|a\rangle$ and $|b\rangle$ independently (domains I and II shown in Fig. 1(c)). This is true provided that the pulse area is sufficient to generate sign changes of the amplitudes: $a(t)$ and $b(t)$, as was shown in Fig. 1(b). Only on the boundary between the one-photon (I) and two-photon (II) regimes, we predict (and detect) interference effects between the one and two photon pathways. **Similar maps can be obtained to study the feasibility of Rabi oscillations in other systems such as atoms or molecules excited to different states** (see our updated outlook in the manuscript). We have calculated such maps for other states in the helium atom, namely the 1s-np Rydberg series, and we have found that the results shown in Fig. 1(c) are universal with some expected shifts of the domains in intensity and time. This suggests that valence-Rydberg short-wavelength Rabi oscillations will be possible to observe in other noble gas atoms.

Finally, we believe that our theory is generalizable to more complex Rabi dynamics with strong photoionization [Ref. X] and autoionization processes [Ref. Y], which would be promising novel directions of exploration for the FEL community for coherent control studies in molecules. We have updated the outlook after taking into account these future aspects.

- I have some quibbles with respect to terminology in the abstract and introduction. Referring to these effects as “quantum optical” is misleading: the effect is fully understood without referring to quantization of the XUV field. The authors should remove the statement that such effects are trademarks of “quantum optics” and simply note that they are fundamental to the physics of atom-field interactions (or similar).

We fully agree that our study did not involve photon counting or fluctuations, traditionally found in quantum optics and while spontaneous emission is mentioned, it is not an actual concern. However, future developments may build on super fluorescence from half integer Rabi cycling at short wavelengths. In this case, quantum optics could be approached concerning the collective effects of many atoms. Here, we change our formulation from

“quantum optics”

to

“fundamental to coherent atom-field interactions”.

Reviewer Reports on the First Revision:

Referees' comments:

Referee #1 (Remarks to the Author):

I appreciate the very detailed and to-the-point answer to all questions raised by the reviewers. I have to admit that I would not have expected the photoionization lifetimes in the present experiment to be as large as now reported in the new panel 1c. Accordingly, I would not have expected the clear and essentially undamped Rabi oscillations now shown in R1 (theory results). These two very convincing clarifications resolve my main criticisms, and also justify the authors' proposal of their method as a new probe for the dynamics. Regarding the possible impact, I fully agree that in general the transition from longer to shorter wavelengths is very desirable, and that it bears potential for many new or improved applications. The present manuscript reports a significant step towards this goal, by demonstrating XUV-induced bound-bound Rabi oscillations for the first time. Nevertheless, I missed some more specific motivation of the experiment in the previous manuscript version, which are now provided in the revised summary/outlook. In this regard, it is also quite remarkable that Fig. 1c shows that there is substantial parameter space left for stronger driving before losses dominate over the coherent dynamics. For these reasons, I recommend publication of the current manuscript version in Nature.

Referee #2 (Remarks to the Author):

This is a resubmission. As already stated in my previous assessment, I would recommend publishing of the results in Nature. The authors have taken actions on all the points of criticism raised by all the referees and the manuscript is ready for publication.

Referee #3 (Remarks to the Author):

I have reviewed the revised manuscript and the rebuttal and i believe that the referees have more than adequately responded to the main points of criticism. The paper should be accepted.